# GROMOV-WASSERSTEIN AUTOENCODERS

**Nao Nakagawa[1], Ren Togo[2], Takahiro Ogawa[2], & Miki Haseyama[2]**
[1] Graduate School of Information Science and Technology, Hokkaido University, Japan
[2] Faculty of Information Science and Technology, Hokkaido University, Japan
`{nakagawa,togo,ogawa,mhaseyama}@lmd.ist.hokudai.ac.jp`

## ABSTRACT

Variational Autoencoder (VAE)-based generative models offer flexible representation learning by incorporating meta-priors, general premises considered beneficial for downstream tasks. However, the incorporated meta-priors often involve ad-hoc model deviations from the original likelihood architecture, causing undesirable changes in their training. In this paper, we propose a novel representation learning method, Gromov-Wasserstein Autoencoders (GWAE), which directly matches the latent and data distributions using the variational autoencoding scheme. Instead of likelihood-based objectives, GWAE models minimize the Gromov-Wasserstein (GW) metric between the trainable prior and given data distributions. The GW metric measures the distance structure-oriented discrepancy between distributions even with different dimensionalities, which provides a direct measure between the latent and data spaces. By restricting the prior family, we can introduce meta-priors into the latent space without changing their objective. The empirical comparisons with VAE-based models show that GWAE models work in two prominent meta-priors, disentanglement and clustering, with their GW objective unchanged.

## 1 INTRODUCTION

One fundamental challenge in unsupervised learning is capturing the underlying low-dimensional structure of high-dimensional data because natural data (*e.g.*, images) lie in low-dimensional manifolds (Carlsson et al., 2008; Bengio et al., 2013). Since deep neural networks have shown their potential for non-linear mapping, representation learning has recently made substantial progress in its applications to high-dimensional and complex data (Kingma & Welling, 2014; Rezende et al., 2014; Hsu et al., 2017; Hu et al., 2017). Learning low-dimensional representations is in mounting demand because the inference of concise representations extracts the essence of data to facilitate various downstream tasks (Thomas et al., 2017; Higgins et al., 2017b; Creager et al., 2019; Locatello et al., 2019a). For obtaining such general-purpose representations, several *meta-priors* have been proposed (Bengio et al., 2013; Tschannen et al., 2018). Meta-priors are general premises about the world, such as disentanglement (Higgins et al., 2017a; Chen et al., 2018; Kim & Mnih, 2018; Ding et al., 2020), hierarchical factors (Vahdat & Kautz, 2020; Zhao et al., 2017; Sønderby et al., 2016), and clustering (Zhao et al., 2018; Zong et al., 2018; Asano et al., 2020).

A prominent approach to representation learning is a deep generative model based on the variational autoencoder (VAE) (Kingma & Welling, 2014). VAE-based models adopt the variational autoencoding scheme, which introduces an inference model in addition to a generative model and thereby offers bidirectionally tractable processes between observed variables (data) and latent variables. In this scheme, the reparameterization trick (Kingma & Welling, 2014) yields representation learning capability since reparameterized latent codes are tractable for gradient computation. The introduction of additional losses and constraints provides further regularization for the training process based on meta-priors. However, controlling representation learning remains a challenging task in VAE-based models owing to the deviation from the original optimization. Whereas the existing VAE-based approaches modify the latent space based on the meta-prior (Kim & Mnih, 2018; Zhao et al., 2017; Zong et al., 2018), their training objectives still partly rely on the evidence lower bound (ELBO). Since the ELBO objective is grounded on variational inference, ad-hoc model modifications cause implicit and undesirable changes, *e.g.*, posterior collapse (Dai et al., 2020) and implicit prior change (Hoffman et al., 2017) in $\beta$-VAE (Higgins et al., 2017a). Under such modifications, it is also unclear whether a

latent representation retains the underlying data structure because VAE models implicitly interpolate data points to form a latent space using noises injected into latent codes by the reparameterization trick (Rezende & Viola, 2018a;b; Aneja et al., 2021).

As another paradigm of variational modeling, the ELBO objective has been reinterpreted from the optimal transport (OT) viewpoint (Tolstikhin et al., 2018). Tolstikhin et al. (2018) have derived a family of generative models called the Wasserstein autoencoder (WAE) by applying the variational autoencoding model to high-dimensional OT problems as the couplings (Appendix A.4 for more details). Despite the OT-based model derivation, the WAE objective is equivalent to that of Info-VAE (Zhao et al., 2019), whose objective consists of the ELBO and the mutual information term. The WAE formulation is derived from the estimation and minimization of the OT cost (Tolstikhin et al., 2018; Arjovsky et al., 2017) between the data distribution and the generative model, *i.e.*, the generative modeling by applying the Wasserstein metric. It furnishes a wide class of models, even when the prior support does not cover the entire variational posterior support. The OT paradigm also applies to existing representation learning approaches originally derived from re-weighting the Kullback-Leibler (KL) divergence term (Gaujac et al., 2021).

Another technique for optimizing the VAE-based ELBO objective called implicit variational inference (IVI) (Huszár, 2017) has been actively researched. While the VAE model has an analytically tractable prior for variational inference, IVI aims at variational inference using *implicit distributions*, in which one can use its sampler instead of its probability density function. A notable approach to IVI is the density ratio estimation (Sugiyama et al., 2012), which replaces the $f$-divergence term in the variational objective with an adversarial discriminator that distinguishes the origin of the samples. For distribution matching, this algorithm shares theoretical grounds with generative models based on the generative adversarial networks (GANs) (Goodfellow et al., 2014; Sønderby et al., 2017), which induces the application of IVI toward the distribution matching in complex and high-dimensional variables, such as images. See Appendix A.6 for more discussions.

In this paper, we propose a novel representation learning methodology, *Gromov-Wasserstein Autoencoder* (GWAE) based on the Gromov-Wasserstein (GW) metric (Mémoli, 2011), an OT-based metric between distributions applicable even with different dimensionality (Mémoli, 2011; Xu et al., 2020; Nguyen et al., 2021). Instead of the ELBO objective, we apply the GW metric objective in the variational autoencoding scheme to directly match the latent marginal (prior) and the data distribution. The GWAE models obtain a latent representation retaining the distance structure of the data space to hold the underlying data information. The GW objective also induces the variational autoencoding to perform the distribution matching of the generative and inference models, despite the OT-based derivation. Under the OT-based variational autoencoding, one can adopt a prior of a GWAE model from a rich class of trainable priors depending on the assumed meta-prior even though the KL divergence from the prior to the encoder is infinite. Our contributions are listed below.

- We propose a novel probabilistic model family GWAE, which matches the latent space to the given unlabeled data via the variational autoencoding scheme. The GWAE models estimate and minimize the GW metric between the latent and data spaces to directly match the latent representation closer to the data in terms of distance structure.

- We propose several families of priors in the form of implicit distributions, adaptively learned from the given dataset using stochastic gradient descent (SGD). The choice of the prior family corresponds to the meta-prior, thereby providing a more flexible modeling scheme for representation learning.

- We conduct empirical evaluations on the capability of GWAE in prominent meta-priors: disentanglement and clustering. Several experiments on image datasets CelebA (Liu et al., 2015), MNIST (LeCun et al., 1998), and 3D Shapes (Burgess & Kim, 2018), show that GWAE models outperform the VAE-based representation learning methods whereas their GW objective is not changed over different meta-priors.

## 2 RELATED WORK

**VAE-based Representation Learning.** VAE (Kingma & Welling, 2014) is a prominent deep generative model for representation learning. Following its theoretical consistency and explicit handling of latent variables, many state-of-the-art representation learning methods are proposed

based on VAE with modification (Higgins et al., 2017a; Chen et al., 2018; Kim & Mnih, 2018; Achille & Soatto, 2018; Kumar et al., 2018; Zong et al., 2018; Zhao et al., 2017; Sønderby et al., 2016; Zhao et al., 2019; Hou et al., 2019; Detlefsen & Hauberg, 2019; Ding et al., 2020). The standard VAE learns an encoder and a decoder with parameters $\phi$ and $\boldsymbol{\theta}$, respectively, to learn a low-dimensional representation in its latent variables $\mathbf{z}$ using a bottleneck layer of the autoencoder. Using data $\mathbf{x} \in p_{\text{data}}(\mathbf{x})$ supported on the data space $\mathcal{X}$, the VAE objective is the ELBO formulated by the following optimization problem:

$$\underset{\boldsymbol{\theta}, \phi}{\text{maximize}} \quad \mathbb{E}_{p_{\text{data}}(\mathbf{x})} \left[ \mathbb{E}_{q_\phi(\mathbf{z}|\mathbf{x})} \left[ \log p_{\boldsymbol{\theta}}(\mathbf{x}|\mathbf{z}) \right] - D_{\text{KL}}(q_\phi(\mathbf{z}|\mathbf{x}) \| \pi(\mathbf{z})) \right], \tag{1}$$

where the encoder $q_\phi(\mathbf{z}|\mathbf{x})$ and decoder $p_{\boldsymbol{\theta}}(\mathbf{x}|\mathbf{z})$ are parameterized by neural networks, and the prior $\pi(\mathbf{z})$ is postulated before training. The first and second terms (called the *reconstruction term* and the *KL term*, respectively) in Eq. (1) are in a trade-off relationship (Tschannen et al., 2018). This implies that learning is guided to autoencoding by the reconstruction term while matching the distribution of latent variables to the pre-defined prior using the KL term.

**Implicit Variational Inference.** IVI solves the variational inference problem using implicit distributions (Huszár, 2017). A major approach to IVI is density ratio estimation (Sugiyama et al., 2012), in which the ratio between probability distribution functions is estimated using a discriminator instead of their closed-form expression. Since IVI-based and GAN-based models share density ratio estimation mechanisms in distribution matching (Sønderby et al., 2017), the combination of VAEs and GANs has been actively studied, especially from the aspect of the matching of implicit distributions. The successful results achieved by GAN-based models in high-dimensional data, such as natural images, have propelled an active application and research of IVI in unsupervised learning (Larsen et al., 2016; Makhzani, 2018).

**Optimal Transport.** The OT cost is used as a measure of the difference between distributions supported on high-dimensional space using SGD (Arjovsky et al., 2017; Tolstikhin et al., 2018; Gaujac et al., 2021). This provides the Wasserstein metric for the discrepancy between distributions. For a constant $\xi \geq 1$, the $\xi$-Wasserstein metric between distributions $r$ and $s$ is defined as

$$W_\xi(r, s) = \left( \inf_{\gamma \in \mathcal{P}(r(\mathbf{x}), s(\mathbf{x}'))} \mathbb{E}_{\gamma(\mathbf{x}, \mathbf{x}')} \left[ d^\xi(\mathbf{x}, \mathbf{x}') \right] \right)^{1/\xi}, \tag{2}$$

where $\mathbf{x}$ denotes the random variable in which the distributions $r$ and $s$ are defined, and $\mathcal{P}(r(\mathbf{x}), s(\mathbf{x}'))$ denotes the set consisting of all couplings whose $\mathbf{x}$-marginal is $r(\mathbf{x})$ and whose $\mathbf{x}'$-marginal is $s(\mathbf{x}')$. Owing to the difficulty of computing the exact infimum in Eq. (2) for high-dimensional, large-scale data, several approaches try to minimize the estimated $\xi$-Wasserstein metric using neural networks and SGD (Tolstikhin et al., 2018; Arjovsky et al., 2017). The form in Eq. (2) is the primal form of the Wasserstein metric, particularly compared with its dual form for the case of $\xi = 1$ (Arjovsky et al., 2017). The two prominent approaches for the OT in high-dimensional, complex large-scale data are: (i) minimizing the primal form using a probabilistic autoencoder (Tolstikhin et al., 2018), and (ii) adversarially optimizing the dual form using a generator-critic pair (Arjovsky et al., 2017).

**Wasserstein Autoencoder (WAE).** WAE (Tolstikhin et al., 2018) is a family of generative models whose autoencoder estimates and minimizes the primal form of the Wasserstein metric between the generative model $p_{\boldsymbol{\theta}}(\mathbf{x})$ and the data distribution $p_{\text{data}}(\mathbf{x})$ using SGD in the variational autoencoding settings, *i.e.*, the VAE model architecture (Kingma & Welling, 2014). This primal-based formulation induces a representation learning methodology from the OT viewpoint because the WAE objective is equivalent to that of InfoVAE (Zhao et al., 2019), which learns the variational autoencoding model by retaining the mutual information of the probabilistic encoder.

**Kantorovich-Rubinstein Duality.** The Wasserstein GAN models (Arjovsky et al., 2017) adopt an objective based on the 1-Wasserstein metric between the generative model $p_{\boldsymbol{\theta}}(\mathbf{x})$ and data distribution $p_{\text{data}}(\mathbf{x})$. This objective is estimated using the Kantorovich-Rubinstein duality (Villiani, 2009; Arjovsky et al., 2017), which holds for the 1-Wasserstein as

$$W_1(r, s) = \sup_{f: \text{1-Lipschitz}} \mathbb{E}_{r(\mathbf{x})} \left[ f(\mathbf{x}) \right] - \mathbb{E}_{s(\mathbf{x})} \left[ f(\mathbf{x}) \right]. \tag{3}$$

To estimate this function $f$ using SGD, a 1-Lipschitz neural network called a *critic* is introduced, as with a discriminator in the GAN-based models. The training process using mini-batches is

adversarially conducted, *i.e.*, by repeating updates of the critic parameters and the generative parameters alternatively. During this process, the critic maximizes the objective in Eq. (3) to approach the supremum, whereas the generative model minimizes the objective for the distribution matching $p_{\boldsymbol{\theta}}(\mathbf{x}) \approx p_{\mathrm{data}}(\mathbf{x})$.

## 3 PROPOSED METHOD

Our GWAE models minimize the OT cost between the data and latent spaces, based on generative modeling in the variational autoencoding. GWAE models learn representations by matching the distance structure between the latent and data spaces, instead of likelihood maximization.

### 3.1 OPTIMAL TRANSPORT BETWEEN SPACES

Although the OT problem induces a metric between probability distributions, its application is limited to distributions sharing one sample space. The GW metric (Mémoli, 2011) measures the discrepancy between metric measure spaces using the OT of distance distributions. A metric measure space consists of a sample space, metric, and probability measure. Given a pair of different metric spaces, *i.e.*, sample spaces and metrics, the GW metric measures the discrepancy between probability distributions supported on the spaces. In terms of the GW metric, two distributions are considered to be equal if there is an isometric mapping between their supports (Sturm, 2012; Sejourne et al., 2021). For a constant $\rho \geq 1$, the formulation of the $\rho$-GW metric between probability distributions $r(\mathbf{x})$ supported on a metric space $(\mathcal{X}, d_{\mathcal{X}})$ and $s(\mathbf{z})$ supported on $(\mathcal{Z}, d_{\mathcal{Z}})$ is given by

$$GW_{\rho}(r, s) := \left( \inf_{\gamma \in \mathcal{P}(r(\mathbf{x}), s(\mathbf{z}))} \mathbb{E}_{\gamma(\mathbf{x}, \mathbf{z})} \mathbb{E}_{\gamma(\mathbf{x}', \mathbf{z}')} \left[ |d_{\mathcal{X}}(\mathbf{x}, \mathbf{x}') - d_{\mathcal{Z}}(\mathbf{z}, \mathbf{z}')|^{\rho} \right] \right)^{1/\rho}, \quad (4)$$

where $\mathcal{P}(r(\mathbf{x}), s(\mathbf{z}))$ denotes the set of all couplings with $r(\mathbf{x})$ as $\mathbf{x}$-marginal and $s(\mathbf{z})$ as $\mathbf{z}$-marginal. The metrics $d_{\mathcal{X}}$ and $d_{\mathcal{Z}}$ are the metrics in the spaces $\mathcal{X}$ and $\mathcal{Z}$, respectively.

### 3.2 APPLICATION TO REPRESENTATION LEARNING: GROMOV-WASSERSTEIN AUTOENCODER

In this work, we propose a novel GWAE modeling methodology based on the GW metric for distance structure modeling in the variational autoencoding formulation. The objectives of generative models typically aim for distribution matching in the data space, *e.g.*, the likelihood (Kingma & Welling, 2014) and the Jensen-Shannon divergence (Goodfellow et al., 2014). The GWAE objective differs from these approaches and aims to directly match the latent and data distributions based on their distance structure.

#### 3.2.1 MODEL SETTINGS: VARIATIONAL AUTOENCODING

Given an $N$-sized set of data points $\{\mathbf{x}_i\}_{i=1}^N$ supported on a data space $\mathcal{X}$, representation learning aims to build a latent space $\mathcal{Z}$ and obtain mappings between both the spaces. For numerical computation, we postulate that the spaces $\mathcal{X}$ and $\mathcal{Z}$ respectively have tractable metrics $d_{\mathcal{X}}$ and $d_{\mathcal{Z}}$ such as the Euclidean distance (see Appendix B.1 for details), and let $M, L \in \mathbb{N} \setminus \{0\}$, $\mathcal{X} \subseteq \mathbb{R}^M$, and $\mathcal{Z} \subseteq \mathbb{R}^L$. We mention the bottleneck case $M \gg L$ similarly to the existing representation learning methods (Kingma & Welling, 2014; Higgins et al., 2017a; Kim & Mnih, 2018) because the data space $\mathcal{X}$ is typically an $L$-dimensional manifold (Carlsson et al., 2008; Bengio et al., 2013).

We construct a model with a trainable latent prior $\pi_{\boldsymbol{\theta}}(\mathbf{z})$ to approach the data distribution $p_{\mathrm{data}}(\mathbf{x})$ in terms of distance structure. Following the standard VAE (Kingma & Welling, 2014), we consider a generative model $p_{\boldsymbol{\theta}}(\mathbf{x}, \mathbf{z})$ with parameters $\boldsymbol{\theta}$ and an inference model $q_{\boldsymbol{\phi}}(\mathbf{x}, \mathbf{z})$ with parameters $\boldsymbol{\phi}$. The generation process consists of the prior $\pi_{\boldsymbol{\theta}}(\mathbf{z})$ and a decoder $p_{\boldsymbol{\theta}}(\mathbf{x}|\mathbf{z})$ parameterized with neural networks. Since the inverted generation process $p_{\boldsymbol{\theta}}(\mathbf{z}|\mathbf{x}) = \pi_{\boldsymbol{\theta}}(\mathbf{z}) p_{\boldsymbol{\theta}}(\mathbf{x}|\mathbf{z}) / p_{\boldsymbol{\theta}}(\mathbf{x})$ is intractable in this scheme, an encoder $q_{\boldsymbol{\phi}}(\mathbf{z}|\mathbf{x}) \approx p_{\boldsymbol{\theta}}(\mathbf{z}|\mathbf{x})$ is instead established using neural networks for parameterization. Thus, the generative $p_{\boldsymbol{\theta}}(\mathbf{x}, \mathbf{z})$ and inference $q_{\boldsymbol{\phi}}(\mathbf{x}, \mathbf{z})$ models are defined as

$$p_{\boldsymbol{\theta}}(\mathbf{x}, \mathbf{z}) = \pi_{\boldsymbol{\theta}}(\mathbf{z}) p_{\boldsymbol{\theta}}(\mathbf{x}|\mathbf{z}), \qquad q_{\boldsymbol{\phi}}(\mathbf{x}, \mathbf{z}) = p_{\mathrm{data}}(\mathbf{x}) q_{\boldsymbol{\phi}}(\mathbf{z}|\mathbf{x}). \qquad (5)$$

The empirical $\hat{p}_{\mathrm{data}}(\mathbf{x}) = 1/N \sum_{i=1}^N \delta(\mathbf{x} - \mathbf{x}_i)$ is used for the estimation of $p_{\mathrm{data}}(\mathbf{x})$. A Dirac decoder and a diagonal Gaussian encoder are used to alleviate deviations from the data manifold as in Tolstikhin et al. (2018) (see Appendix B.1 for these details and formulations).

### 3.2.2 OPTIMAL TRANSPORT OBJECTIVE

Here, we focus on the latent space $\mathcal{Z}$ to transfer the underlying data structure to the latent space. This highlights the main difference between the GWAE and the existing generative approaches. The training objective of GWAE is the GW metric between the metric measure spaces $(\mathcal{X}, d_{\mathcal{X}}, p_{\text{data}}(\mathbf{x}))$ and $(\mathcal{Z}, d_{\mathcal{Z}}, \pi_{\boldsymbol{\theta}}(\mathbf{z}))$ as

$$\underset{\boldsymbol{\theta}}{\text{minimize}} \quad GW_{\rho}(p_{\text{data}}(\mathbf{x}), \pi_{\boldsymbol{\theta}}(\mathbf{z}))^{\rho}, \tag{6}$$

where $\rho \geq 1$ is a constant, and we adopt $\rho = 1$ to alleviate the effect of outlier samples distant from the isometry for training stability. Computing the exact GW value is difficult owing to the high dimensionality of both $\mathbf{x}$ and $\mathbf{z}$. Hence, we estimate and minimize the GW metric using the variational autoencoding scheme, which captures the latent factors of complex data in a stable manner. We recast the GW objective into a main GW estimator $\mathcal{L}_{GW}$ with three regularizations: a reconstruction loss $\mathcal{L}_W$, a joint dual loss $\mathcal{L}_D$, and an entropy regularization $\mathcal{R}_{\mathcal{H}}$.

**Estimated GW metric $\mathcal{L}_{GW}$.** We use the generative model $p_{\boldsymbol{\theta}}(\mathbf{x}, \mathbf{z})$ as the coupling of Eq. (6) similarly to the WAE (Tolstikhin et al., 2018) methodology. The main loss $\mathcal{L}_{GW}$ estimates the GW metric as:

$$\underset{\boldsymbol{\theta}}{\text{minimize}} \quad \mathcal{L}_{GW} := \mathbb{E}_{p_{\boldsymbol{\theta}}(\mathbf{x}, \mathbf{z})} \mathbb{E}_{p_{\boldsymbol{\theta}}(\mathbf{x}', \mathbf{z}')} \left[ |d_{\mathcal{X}}(\mathbf{x}, \mathbf{x}') - C d_{\mathcal{Z}}(\mathbf{z}, \mathbf{z}')|^{\rho} \right], \tag{7}$$

$$\text{subject to} \quad p_{\text{data}}(\mathbf{x}) = p_{\boldsymbol{\theta}}(\mathbf{x}), \tag{8}$$

where $C$ is a trainable scale constant to cancel out the scale degree of freedom, and $p_{\boldsymbol{\theta}}(\mathbf{x})$ denotes the marginal $p_{\boldsymbol{\theta}}(\mathbf{x}) = \int_{\mathcal{Z}} p_{\boldsymbol{\theta}}(\mathbf{x}, \mathbf{z}) d\mathbf{z}$.

**WAE-based $\mathcal{X}$-marginal condition $\mathcal{L}_W$.** To obtain a numerical solution with stable training, Tolstikhin et al. (2018) relax the $\mathcal{X}$-matching condition of Eq. (8) into $\xi$-Wasserstein minimization ($\xi \geq 1$) using the variational autoencoding coupling. The WAE methodology (Tolstikhin et al., 2018) uses the inference model $q_{\boldsymbol{\phi}}(\mathbf{x}, \mathbf{z})$ to formulate the $\xi$-Wasserstein minimization as the reconstruction loss $\mathcal{L}_W$ with a $\mathcal{Z}$-matching condition as:

$$\underset{\boldsymbol{\theta}, \boldsymbol{\phi}}{\text{minimize}} \quad \mathcal{L}_W := \mathbb{E}_{q_{\boldsymbol{\phi}}(\mathbf{x}, \mathbf{z})} \mathbb{E}_{p_{\boldsymbol{\theta}}(\mathbf{x}'|\mathbf{z})} \left[ d_{\mathcal{X}}(\mathbf{x}, \mathbf{x}') \right], \tag{9}$$

$$\text{subject to} \quad q_{\boldsymbol{\phi}}(\mathbf{z}) = \pi_{\boldsymbol{\theta}}(\mathbf{z}). \tag{10}$$

where $d_{\mathcal{X}}$ is a distance function based on the $L_{\xi}$ metric. We adopt the settings $\xi = 2$ to retain the conventional Gaussian reconstruction loss.

**Merged sufficient condition $\mathcal{L}_D$.** We merge the marginal coupling conditions of Eq. (8) and Eq. (10) into the joint $\mathcal{X} \times \mathcal{Z}$-matching sufficient condition $p_{\boldsymbol{\theta}}(\mathbf{x}, \mathbf{z}) = q_{\boldsymbol{\phi}}(\mathbf{x}, \mathbf{z})$ to attain bidirectional inferences while preserving the stability of autoencoding. Since such joint distribution matching can also be relaxed into the minimization of $W_1(q_{\boldsymbol{\phi}}(\mathbf{x}, \mathbf{z}), p_{\boldsymbol{\theta}}(\mathbf{x}, \mathbf{z}))$, this condition is satisfied by minimizing the Kantorovich-Rubinstein duality introduced by Arjovsky et al. (2017) as in Eq. (3). Practically, a 1-Lipschitz neural network (critic) $f_{\boldsymbol{\psi}}$ estimates the supremum of Eq. (3), and the main model minimizes this estimated supremum as:

$$\underset{\boldsymbol{\theta}, \boldsymbol{\phi}}{\text{minimize}} \underset{\boldsymbol{\psi}}{\text{maximize}} \quad \mathcal{L}_D := \mathbb{E}_{q_{\boldsymbol{\phi}}(\mathbf{x}, \mathbf{z})} \left[ f_{\boldsymbol{\psi}}(\mathbf{x}, \mathbf{z}) \right] - \mathbb{E}_{p_{\boldsymbol{\theta}}(\mathbf{x}, \mathbf{z})} \left[ f_{\boldsymbol{\psi}}(\mathbf{x}, \mathbf{z}) \right], \tag{11}$$

where $\boldsymbol{\psi}$ is the critic parameters. To satisfy the 1-Lipschitz constraint, the critic $f_{\boldsymbol{\psi}}$ is implemented with techniques such as spectral normalization (Miyato et al., 2018) and gradient penalty (Gulrajani et al., 2017) (see Appendix B.3 for the details of the gradient penalty loss).

**Entropy regularization $\mathcal{R}_{\mathcal{H}}$.** We further introduce the entropy regularization $\mathcal{R}_{\mathcal{H}}$ using the inference entropy to avoid degenerate solutions in which the encoder $q_{\boldsymbol{\phi}}(\mathbf{z}|\mathbf{x})$ becomes Dirac and deterministic for all data points. In such degenerate solutions, the latent representation simply becomes a look-up table because such a point-to-point encoder maps the set of data points into a set of latent code points with measure zero (Hoffman et al., 2017; Dai et al., 2018), causing overfitting into the empirical data distribution. An effective way to avoid it is a regularization with the inference entropy $\mathcal{H}_q$ of the latent variables $\mathbf{z}$ conditioned on data $\mathbf{x}$ as

$$\mathcal{R}_{\mathcal{H}} := \mathcal{H}_q(\mathbf{z}|\mathbf{x}) = \mathbb{E}_{q_{\boldsymbol{\phi}}(\mathbf{x}, \mathbf{z})} \left[ -\log q_{\boldsymbol{\phi}}(\mathbf{z}|\mathbf{x}) \right]. \tag{12}$$

Since the conditioned entropy $\mathcal{H}_q(\mathbf{z}|\mathbf{x})$ diverges to negative infinity in the degenerate solutions, the regularization term $-\mathcal{R}_{\mathcal{H}}$ facilitates the probabilistic learning of GWAE models.

**Stochastic Training with Single Estimated Objective.** Applying the Lagrange multiplier method to the aforementioned constraints, we recast the GW metric of Eq. (6) into a single objective $\mathcal{L}$ with multipliers $\lambda_W$, $\lambda_D$, and $\lambda_{\mathcal{H}}$ as

$$\underset{\boldsymbol{\theta}, \boldsymbol{\phi}}{\text{minimize}} \, \underset{\boldsymbol{\psi}}{\text{maximize}} \quad \mathcal{L} := \mathcal{L}_{GW} + \lambda_W \mathcal{L}_W + \lambda_D \mathcal{L}_D - \lambda_{\mathcal{H}} \mathcal{R}_{\mathcal{H}}. \tag{13}$$

One efficient solution to optimize this objective is using the mini-batch gradient descent in alternative steps (Goodfellow et al., 2014; Arjovsky et al., 2017), which we can conduct in automatic differentiation packages, such as PyTorch (Paszke et al., 2019). One step of mini-batch descent is the minimization of the total objective $\mathcal{L}$ in Eq. (13), and the other step is the maximization of the critic objective $\mathcal{L}_D$ in Eq. (11). By alternatively repeating these steps, the critic estimates the Wasserstein metric using the expected potential difference $\mathcal{L}_D$ (Arjovsky et al., 2017). Although the objective in Eq. (13) involves three auxiliary regularizations including an adversarial term, the GWAE model can be efficiently optimized because the adversarial mechanism and the variational autoencoding scheme share the goal of distribution matching $p_{\boldsymbol{\theta}}(\mathbf{x}, \mathbf{z}) \approx q_{\boldsymbol{\phi}}(\mathbf{x}, \mathbf{z})$ (see Appendix C.5 for more details).

### 3.2.3 PRIOR BY SAMPLING

GWAE models apply to the cases in which the prior $\pi_{\boldsymbol{\theta}}(\mathbf{z})$ takes the form of an implicit distribution with a sampler. An implicit distribution $\pi_{\boldsymbol{\theta}}(\mathbf{z})$ provides its sampler $\mathbf{z} \sim \pi_{\boldsymbol{\theta}}(\mathbf{z})$ while a closed-form expression of the probability density function is not available. The adversarial algorithm of GWAE handles such cases and enables a wide class of priors to provide meta-prior-based inductive biases for unsupervised representation learning, *e.g.*, for disentanglement (Locatello et al., 2019b; 2020). Note that the GW objective in Eq. (6) becomes a constant function in non-trainable prior cases.

**Neural Prior (NP).** A straightforward way to build a differentiable sampler of a trainable prior is using a neural network to convert noises. The prior of the latent variables $\mathbf{z}$ is defined via sampling using a neural network $g_{\boldsymbol{\theta}} : \mathbb{R}^L \to \mathbb{R}^L$ with parameters $\boldsymbol{\theta}$ (see Appendix B.2 for its formulation). Notably, the neural network $g_{\boldsymbol{\theta}}$ need not be invertible unlike Normalizing Flow (Rezende & Mohamed, 2015) since the prior is defined as an implicit distribution not requiring a push-forward measure.

**Factorized Neural Prior (FNP).** For disentanglement, we can constitute a factorized prior using an element-wise independent neural network $\tilde{g}_{\boldsymbol{\theta}} = \{\tilde{g}_{\boldsymbol{\theta}}^{(i)}\}_{i=1}^L$ (see Appendix B.2 for its formulation). Such factorized priors can be easily implemented utilizing the 1-dimensional grouped convolution (Krizhevsky et al., 2012).

**Gaussian Mixture Prior (GMP).** For clustering structure, we construct a class of Gaussian mixture priors. Given that the prior contains $K$ components, the $k$-th component is parameterized using the weights $w_k$, means $\mathbf{m}_k \in \mathbb{R}^L$, and square-root covariances $\mathbf{M}_k \in \mathbb{R}^{L \times L}$ as

$$\pi_{\boldsymbol{\theta}}(\mathbf{z}) = \sum_{k=1}^K w_k \mathcal{N}(\mathbf{z} | \mathbf{m}_k, \mathbf{M}_k \mathbf{M}_k^\mathsf{T}), \tag{14}$$

where the weights $\{w_k\}_{k=1}^K$ are normalized as $\sum_{k=1}^K w_k = 1$. To sample from a prior of this class, one randomly chooses a component $k$ from the $K$-way categorical distribution with probabilities $(w_1, w_2, \ldots, w_k)$ and draws a sample $\mathbf{z}$ as follows:

$$\mathbf{z} = \mathbf{m}_k + \mathbf{M}_k \boldsymbol{\epsilon}, \qquad \boldsymbol{\epsilon} \sim \mathcal{N}(\mathbf{0}, \mathbf{I}_L), \tag{15}$$

where $\mathbf{0}$ and $\mathbf{I}_n$ denote the zero vector and the $n$-sized identity matrix, respectively. In this class of priors, the set of trainable parameters consists of $\{(w_k, \mathbf{m}_k, \mathbf{M}_k)\}_{k=1}^K$. Note that this parameterization can be easily implemented in differentiable programming frameworks because $\mathbf{M}_k \mathbf{M}_k^\mathsf{T}$ is positive semidefinite for any $\mathbf{M}_k \in \mathbb{R}^{L \times L}$.

## 4 EXPERIMENTS

We investigated the wide capability of the GWAE models for learning representations based on meta-priors.[1] We evaluated GWAEs in two principal meta-priors: disentanglement and clustering.

---

[1] In the tables of the quantitative evaluations, $\uparrow$ and $\downarrow$ indicate scores in which higher and lower values are better, respectively.

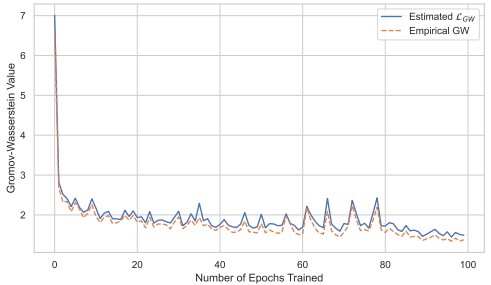 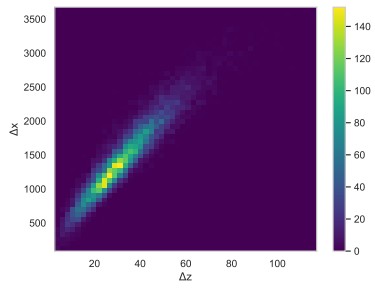

(a) The estimation of the GW metric in each epoch.    (b) The isometry in GWAE.

Figure 1: The estimation and minimization of the GW metric. This trial of training is conduced in GWAE (NP, $\lambda_D$=1, $\lambda_W$=1, $\lambda_{\mathcal{H}}$=1) using the MNIST (LeCun et al., 1998) dataset. (a) The curves show the GW values estimated by the loss term $\mathcal{L}_{GW}$ (solid, blue) and the empirical GW computed by the POT package (Flamary et al., 2021) (dashed, orange). The values are computed using the validation set. (b) The axes $\Delta x = d_{\mathcal{X}}(\mathbf{x}, \mathbf{x}')$ (vertical) and $\Delta z = d_{\mathcal{Z}}(\mathbf{z}, \mathbf{z}')$ (horizontal) respectively denote the difference in the data and latent spaces between generated samples $(\mathbf{x}, \mathbf{z}), (\mathbf{x}', \mathbf{z}') \sim p_{\boldsymbol{\theta}}(\mathbf{x}, \mathbf{z})$. The histogram contains 10,000 generated sample pairs.

To validate the effectiveness of GWAE on different tasks for each meta-prior, we conducted each experiment in corresponding experimental settings. We further studied their autoencoding and generation for the inspection of general capability.

## 4.1 Experimental Settings

We compared the GWAE models with existing representation learning methods (see Appendix A for the details of the compared methods). For the experimental results in this section, we used four visual datasets: CelebA (Liu et al., 2015), MNIST (LeCun et al., 1998), 3D Shapes (Burgess & Kim, 2018), and Omniglot (Lake et al., 2015) (see Appendix C.1 for dataset details). For quantitative evaluations, we selected hyperparameters from $\lambda_W \in [10^0, 10^1]$, $\lambda_D \in [10^0, 10^1]$, and $\lambda_{\mathcal{H}} \in [10^{-4}, 10^0]$ using their performance on the validation set. For fair comparisons, we trained the networks with a consistent architecture from scratch in all the methods (see Appendix C.2 for architecture details).

## 4.2 Gromov-Wasserstein Estimation and Minimization

We validated the estimation and minimization of the GW metric in Fig. 1. First, to validate the estimation of the GW metric, we compared the GW metric estimated in GWAE and the empirical GW value computed in the conventional method in Fig. 1a. Against the GWAE models estimating the GW metric as in Eq. (7), the empirical GW values are computed by the standard OT framework POT (Flamary et al., 2021). Although the estimated $\mathcal{L}_{GW}$ is slightly higher than the empirical values, the curves behave in a very similar manner during the entire training process. This result supports that the GWAE model successfully estimated the GW values and yielded their gradients to proceed with the distribution matching between the data and latent spaces. Second, to validate the minimization of the GW metric, we show the histogram of the differences of generated samples in the data and latent space in Fig. 1b. The isometry of generated samples is attained if the generative coupling $p_{\boldsymbol{\theta}}(\mathbf{x}, \mathbf{z})$ attains the infimum in Eq. (4). This histogram result shows that the generative model $p_{\boldsymbol{\theta}}(\mathbf{x}, \mathbf{z})$ acquired nearly-isometric latent embedding, and suggests that the GW metric was successfully minimized although the objective of Eq. (13) contains three regularization loss terms (refer to Appendix C.8 for ablation studies, and Appendix C.4 for comparisons). These two experimental results support that the GWAE models successfully estimated and optimized the GW objective.

## 4.3 Learning Representations Based on Meta-Priors

**Disentanglement.** We investigated the disentanglement of representations obtained using GWAE models and compared them with conventional VAE-based disentanglement methods. Since the element-wise independence in the latent space is postulated as a meta-prior for disentangled representation learning, we used the FNP class for the prior $\pi_{\boldsymbol{\theta}}(\mathbf{z})$. Considering practical applications with

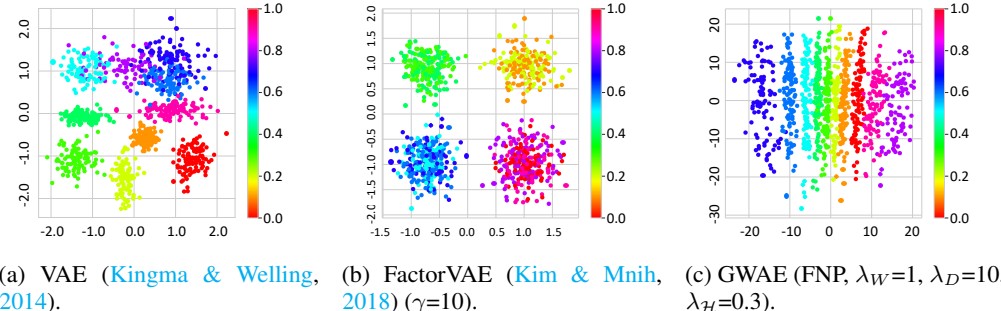

(a) VAE (Kingma & Welling, 2014).

(b) FactorVAE (Kim & Mnih, 2018) ($\gamma$=10).

(c) GWAE (FNP, $\lambda_W$=1, $\lambda_D$=10, $\lambda_{\mathcal{H}}$=0.3).

Figure 2: Comparison of the learned latent spaces in 3D Shapes (Burgess & Kim, 2018) and $L = 16$. The vertical and horizontal axes in the scatter plots respectively represent two of the 16 (= $L$) latent variables with the highest and the second-highest informativeness (Do & Tran, 2020) w.r.t. the *object hue* factor. Note that a single factor value varies along only one axis in a disentangled representation.

Table 1: Quantitative comparison of disentanglement. The reported scores were calculated in 3D Shapes (Burgess & Kim, 2018), and the latent size $L = 16$. Since the latent size $L$ is larger than the number of the ground truth factors, the hyperparameter tuning was based on the validation set DCI-C (Eastwood & Williams, 2018) values. To deal with the probabilistic scores (Zaidi et al., 2021), we reported the ranges for five measurements. The details of the scores are provided in Appendix C.3.

| Model | DCI-C ↑ | DCI-D ↑ | DCI-I ↑ |
|---|---|---|---|
| VAE (Kingma & Welling, 2014) | $0.7734 \pm 0.0004$ | $0.6831 \pm 0.0002$ | $0.9914 \pm 0.0003$ |
| $\beta$-VAE (Higgins et al., 2017a) | $0.8245 \pm 0.0002$ | $0.7328 \pm 0.0002$ | $0.9796 \pm 0.0002$ |
| WAE (Tolstikhin et al., 2018) | $0.8288 \pm 0.0004$ | $\mathbf{0.7544 \pm 0.0004}$ | $0.9959 \pm 0.0001$ |
| $\beta$-TCVAE (Chen et al., 2018) | $0.8347 \pm 0.0003$ | $0.7085 \pm 0.0002$ | $0.9880 \pm 0.0002$ |
| FactorVAE (Kim & Mnih, 2018) | $0.7963 \pm 0.0004$ | $0.7390 \pm 0.0004$ | $0.9961 \pm 0.0002$ |
| DIP-VAE-I (Kumar et al., 2018) | $0.8609 \pm 0.0003$ | $0.6984 \pm 0.0003$ | $0.9961 \pm 0.0001$ |
| DIP-VAE-II (Kumar et al., 2018) | $0.8236 \pm 0.0001$ | $0.7498 \pm 0.0003$ | $0.9957 \pm 0.0002$ |
| GWAE (FNP) | $\mathbf{0.9080 \pm 0.0002}$ | $0.7024 \pm 0.0002$ | $\mathbf{0.9966 \pm 0.0002}$ |

\* The ranges are denoted by (mean) $\pm$ (standard error of the mean).

unknown ground-truth factor, we set relatively large latent size $L$ to avoid the shortage of dimensionality. The qualitative and quantitative results are shown in Fig. 2 and Table 1, respectively. These results support the ability to learn a disentangled representation in complex data. The scatter plots in Fig. 2 suggest that the GWAE model successfully extracted one underlying factor of variation (object hue) precisely along one axis, whereas the standard VAE (Kingma & Welling, 2014) formed several clusters for each value, and FactorVAE (Kim & Mnih, 2018) obtained the factor in quadrants.

**Clustering Structure.** We empirically evaluated the capabilities of capturing clusters using MNIST (LeCun et al., 1998). We compared the GWAE model using GMP with other VAE-based methods considering the out-of-distribution (OoD) detection performance in Fig. 3. We used MNIST images as in-distribution (ID) samples for training and Omniglot (Lake et al., 2015) images as unseen OoD samples. Quantitative results show that the GWAE model successfully extracted the clustering structure, empirically implying the applicability of multimodal priors.

### 4.4 AUTOENCODING MODEL

We additionally studied the autoencoding and generation performance of GWAE models in Table 2 (see Appendix C.7 for qualitative evaluations). Although the distribution matching $p_{\theta}(\mathbf{x}) \approx p_{\text{data}}(\mathbf{x})$ is a collateral condition of Eq. (7), quantitative results show that the GWAE model also favorably compares with existing autoencoding models in terms of generative capacity. This result suggests the substantial capture of the underlying low-dimensional distribution in GWAE models, which can lead to the applications to other types of meta-priors.

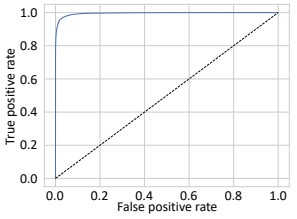 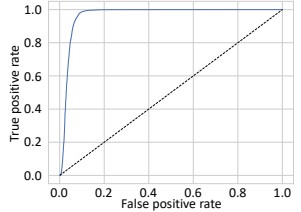 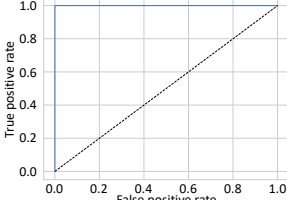

(a) VAE (Kingma & Welling, 2014), AUC=0.9957.

(b) DAGMM (Zong et al., 2018), AUC=0.9654.

(c) GWAE (GMP), AUC=1.0000.

Figure 3: The ROC curves of the OoD detection in MNIST (LeCun et al., 1998) against Omniglot (Lake et al., 2015). We trained these models using MNIST as ID samples and used Omniglot as OoD samples. We upsampled Omniglot to 10,000 samples for data balancing. For the anomaly detection using the latent codes $\mathbf{z}$, we applied the negative log-likelihood energy $-\log \pi(\mathbf{z})$ for VAE and DAGMM, and used the estimated Kantorovich potential $\mathbb{E}_{p_{\boldsymbol{\theta}}(\tilde{\mathbf{x}}|\mathbf{z})}[f_{\boldsymbol{\psi}}(\tilde{\mathbf{x}}, \mathbf{z})]$ for GWAE (see Appendix C.10 for more latent space details).

Table 2: Quantitative comparisons of generation and reconstruction. The FID scores (Heusel et al., 2017) evaluate a random sample set from the generative model $p_{\boldsymbol{\theta}}(\mathbf{x})$ (without using dataset images) against the entire test set, and both consist of an equal number of 19,962 samples. The PSNR scores measure the reconstruction $q_{\boldsymbol{\phi}}(\mathbf{z})p_{\boldsymbol{\theta}}(\mathbf{x}|\mathbf{z})$ using test images (see Appendix C.3 for details). All reported values were computed in CelebA (Liu et al., 2015) with a latent size of $L = 64$. For all the methods, we applied early stopping (patience=10) and hyperparameter tuning using the validation set. The bold and underlined values respectively denote **the best** and the second-best performance in each score.

| Model | | FID $\downarrow$ | PSNR [dB] $\uparrow$ |
|---|---|---|---|
| Baseline | VAE (Kingma & Welling, 2014) | 130.9 | 19.96 |
| KL re-weighting | $\beta$-VAE (Higgins et al., 2017a) | 92.6 | 22.71 |
| | GECO (Rezende & Viola, 2018a) | 162.1 | 21.19 |
| | $\sigma$-VAE (Rybkin et al., 2021) | 53.13* | 20.03 |
| Hierarchical factors | LadderVAE (Sønderby et al., 2016) | 255.6 | 12.35 |
| | VLadderAE (Zhao et al., 2017) | 147.1 | 19.76 |
| OT-based models | WAE (Tolstikhin et al., 2018) | 55* | 22.70 |
| | WVI (Ambrogioni et al., 2018) | 295.0 | 14.45 |
| | SWAE (Kolouri et al., 2019) | 102.2 | 21.85 |
| | RAE (Xu et al., 2020) | 52.20* | 21.34 |
| Trainable priors | VampPrior (Tomczak & Welling, 2018) | 243.8 | 16.23 |
| | 2-Stage VAE (Dai & Wipf, 2019) | **34**\* | 16.15 |
| IVI-based models | VAE-GAN (Larsen et al., 2016) | 111.8 | 19.51 |
| | AVB (Mescheder et al., 2017) | 93.0 | 22.60 |
| | ALI (Dumoulin et al., 2017) | 171.8 | 12.26 |
| Ours | GWAE (NP) | 45.3 | **22.82** |

\* The values are cited from the original papers annotated after the model names.

## 5 CONCLUSION

In this work, we have introduced a novel representation learning method that performs the distance distribution matching between the given unlabeled data and the latent space. Our GWAE model family transfers distance structure from the data space into the latent space in the OT viewpoint, replacing the ELBO objective of variational inference with the GW metric. The GW objective provides a direct measure between the latent and data distribution. Qualitative and quantitative evaluations empirically show the performance of GWAE models in terms of representation learning. In future work, further applications also remain open to various types of meta-priors, such as spherical representations and non-Euclidean embedding spaces.

## REPRODUCIBILITY STATEMENT

We describe the implementation details in Section 4, Appendix B, and Appendix C. The dataset details are provided in Appendix C.1. To ensure reproducibility, our code is available online at https://github.com/ganmodokix/gwae and is provided as the supplementary material.

## ACKNOWLEDGMENTS

This work was partly supported by AMED Grant Number JP21zf0127004 and JSPS KAKENHI Grant Number JP21H03456.

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

## A    DETAILS OF RELATED WORK

For self-containment, we describe VAE-based representation learning methods. As with Section 3, $\mathbf{x}$ and $\mathbf{z}$ denote data and latent variables, respectively, and the data $\mathbf{x}$ are $M$-dimensional and the latent variables $\mathbf{z}$ are $L$-dimensional. Unless otherwise noted, each VAE-based model consists of a generative model $p_{\boldsymbol{\theta}}(\mathbf{x}, \mathbf{z})$ with parameters $\boldsymbol{\theta}$, an inference model $q_{\boldsymbol{\phi}}(\mathbf{x}, \mathbf{z})$ with parameters $\boldsymbol{\phi}$, and a pre-defined (non-trainable) prior $\pi(\mathbf{z})$ as in the standard VAE model architecture.

### A.1    VAE-BASED MODELS WITH ELBO EXTENSION

Utilizing the latent variables of VAE-based models is a prominent approach to representation learning. Several models with extended ELBO-based objectives aim to overcome the shortcomings of the original VAE model, such as posterior collapse. VAE-based models are mainly grounded on the ELBO objective, where we denote the ELBO for the data point $\mathbf{x}$ as

$$\text{ELBO}(\mathbf{x}; \boldsymbol{\theta}, \boldsymbol{\phi}) = \mathbb{E}_{q_{\boldsymbol{\phi}}(\mathbf{z}|\mathbf{x})}\left[\log p_{\boldsymbol{\theta}}(\mathbf{x}|\mathbf{z})\right] - D_{\text{KL}}(q_{\boldsymbol{\phi}}(\mathbf{z}|\mathbf{x}) \| \pi(\mathbf{z})), \tag{16}$$

which is mentioned as the expected objective of the original VAE (Kingma & Welling, 2014) in Eq. (1).

#### A.1.1    $\beta$-VAE

$\beta$-VAE (Higgins et al., 2017a) is a VAE-based model for learning disentangled representations by re-weighting the KL term of the ELBO. Given a KKT multiplier $\beta > 0$, the $\beta$-VAE objective is expressed as

$$\underset{\boldsymbol{\theta}, \boldsymbol{\phi}}{\text{maximize}} \quad \mathbb{E}_{p_{\text{data}}(\mathbf{x})}\left[\mathbb{E}_{q_{\boldsymbol{\phi}}(\mathbf{z}|\mathbf{x})}\left[p_{\boldsymbol{\theta}}(\mathbf{x}|\mathbf{z})\right] - \beta D_{\text{KL}}(q_{\boldsymbol{\phi}}(\mathbf{z}|\mathbf{x}) \| \pi(\mathbf{z}))\right]. \tag{17}$$

The KKT multiplier $\beta$ works as the weight of the regularization to impose a factorized prior (*e.g.*, the standard Gaussian $\mathcal{N}(\mathbf{0}, \mathbf{I}_L)$) on the latent variables. This re-weighting induces the capability of disentanglement in the case of $\beta > 1$; however, a large value of $\beta$ causes posterior collapse, in which the latent variables "forget" the information of the input data.

From the Information Bottleneck (IB) (Tishby et al., 1999) point of view, the $\beta$-VAE objective is re-interpreted as the following optimization problem (Alemi et al., 2018; Achille & Soatto, 2018):

$$\underset{\boldsymbol{\theta}, \boldsymbol{\phi}}{\text{maximize}} \quad I_{\boldsymbol{\phi}}(\mathbf{z}; \mathbf{y}) \tag{18}$$

$$\text{subject to} \quad I_{\boldsymbol{\phi}}(\mathbf{z}; \mathbf{x}) \leq I_c, \tag{19}$$

where $I_c$ is a bottleneck capacity, $\mathbf{y}$ is a task to be estimated, and $I_{\boldsymbol{\phi}}(\cdot; \cdot)$ denotes the mutual information on the inference model. Introducing the Lagrange multiplier $\beta$, the IB problem is given as

$$\underset{\boldsymbol{\theta}, \boldsymbol{\phi}}{\text{maximize}} \quad I_{\boldsymbol{\phi}}(\mathbf{z}; \mathbf{y}) - \beta I_{\boldsymbol{\phi}}(\mathbf{z}; \mathbf{x}). \tag{20}$$

Alemi et al. (2018) have given the lower bound of this IB objective as

$$I_{\boldsymbol{\phi}}(\mathbf{z}; \mathbf{y}) - \beta I_{\boldsymbol{\phi}}(\mathbf{z}; \mathbf{x}) \geq \underbrace{\mathbb{E}_{p_{\text{data}}(\mathbf{y})q_{\boldsymbol{\phi}}(\mathbf{z}|\mathbf{y})}\left[\log p_{\boldsymbol{\theta}}(\mathbf{y}|\mathbf{z})\right] - \mathcal{H}(\mathbf{y})}_{\text{The lower bound of } I_{\boldsymbol{\phi}}(\mathbf{z};\mathbf{y})} - \beta \underbrace{\mathbb{E}_{p_{\text{data}}(\mathbf{x})}\left[D_{\text{KL}}(q_{\boldsymbol{\phi}}(\mathbf{z}|\mathbf{x}) \| \pi(\mathbf{z}))\right]}_{\text{The upper bound of } I_{\boldsymbol{\phi}}(\mathbf{z};\mathbf{x})},$$

$$\tag{21}$$

where the task entropy $\mathcal{H}(\mathbf{y})$ is independent of the parameters $\boldsymbol{\theta}$ and $\boldsymbol{\phi}$. The autoencoding task $\mathbf{y} = \mathbf{x}$ gives the objective equivalent to that of the original VAE. This IB-based formulation of the $\beta$-VAE objective implies that the larger value of the multiplier $\beta$ guides the training process to minimize the mutual information $I_{\boldsymbol{\phi}}(\mathbf{z}; \mathbf{x})$ to make the encoder forget the input data, *i.e.*, to cause posterior collapse.

### A.1.2 FACTORVAE

FactorVAE (Kim & Mnih, 2018) is a state-of-the-art disentanglement method that minimizes the Total Correlation (TC) of the aggregated posterior $q_{\phi}(\mathbf{z}) = \mathbb{E}_{p_{\mathrm{data}}(\mathbf{x})}[q_{\phi}(\mathbf{z}|\mathbf{x})]$ in addition to the original ELBO objective. The TC is expressed as the KL divergence between a distribution and its factorized counterpart. In the FactorVAE case, the TC of the aggregated posterior is the KL divergence from the factorized aggregated posterior $\bar{q}_{\phi}(\mathbf{z}) = \prod_{i=1}^{L} q_{\phi}(z_i)$ to the aggregated posterior $q_{\phi}(\mathbf{z})$. The training objective of FactorVAE is the weighted sum of the ELBO and the TC term as

$$\underset{\boldsymbol{\theta},\boldsymbol{\phi}}{\mathrm{maximize}} \quad \mathrm{ELBO}(\mathbf{x};\boldsymbol{\theta},\boldsymbol{\phi}) - \gamma\mathrm{TC}(q_{\phi}(\mathbf{z})), \tag{22}$$

where $\mathrm{TC}(\mathbf{z})$ denotes the TC of the latent variables $\mathbf{z}$ defined as

$$\mathrm{TC}(\mathbf{z}) = D_{\mathrm{KL}}(q_{\phi}(\mathbf{z})\|\bar{q}_{\phi}(\mathbf{z})) \tag{23}$$

$$= \mathbb{E}_{q_{\phi}(\mathbf{z})}\left[\log\frac{\mathrm{Disc}(\mathbf{z})}{1 - \mathrm{Disc}(\mathbf{z})}\right]. \tag{24}$$

In Eq. (24), $\mathrm{Disc}(\mathbf{z})$ denotes a discriminator to estimate the TC term by density ratio estimation (Sugiyama et al., 2012) as

$$\mathrm{Disc}(\mathbf{z}) = \arg\max_{f:\mathcal{Z}\rightarrow[0,1]} \mathbb{E}_{q_{\phi}(\mathbf{z})}\left[\log f(\mathbf{z})\right] + \mathbb{E}_{\bar{q}_{\phi}(\mathbf{z})}\left[\log(1 - f(\mathbf{z}))\right]. \tag{25}$$

Practically, the discriminator is estimated using SGD in parallel using samples from $\bar{q}_{\phi}(\mathbf{z})$ by permuting the latent codes along the batch dimension independently in each latent variable.

### A.1.3 INFOVAE

InfoVAE (Zhao et al., 2019) is an extension of VAE to prevent posterior collapse by the retention of data information in the latent variables. The InfoVAE objective is the sum of the ELBO and the inference model mutual information $I_{\phi}$ in Eq. (19). To this end, the following maximization problem is solved via SGD:

$$\underset{\boldsymbol{\theta},\boldsymbol{\phi}}{\mathrm{maximize}} \quad \mathbb{E}_{p_{\mathrm{data}}(\mathbf{x})}\left[\mathrm{ELBO}(\mathbf{x};\boldsymbol{\theta},\boldsymbol{\phi})\right] + I_{\phi}(\mathbf{x};\mathbf{z}) \tag{26}$$

$$= \mathbb{E}_{p_{\mathrm{data}}(\mathbf{x})}\mathbb{E}_{q_{\phi}(\mathbf{z}|\mathbf{x})}\left[p_{\boldsymbol{\theta}}(\mathbf{x}|\mathbf{z})\right] - D_{\mathrm{KL}}(q_{\phi}(\mathbf{z})\|\pi(\mathbf{z})) \tag{27}$$

The main difference between the VAE and InfoVAE objectives is using the regularization term $D_{\mathrm{KL}}(q_{\phi}(\mathbf{z})\|\pi(\mathbf{z}))$ instead of the original VAE regularization $D_{\mathrm{KL}}(q_{\phi}(\mathbf{z}|\mathbf{x})\|\pi(\mathbf{z}))$. The original KL term becomes zero if all the data points are encoded into the standard Gaussian $\mathcal{N}(\mathbf{0},\mathbf{I}_L)$ to cause posterior collapse. The InfoVAE KL term $D_{\mathrm{KL}}(q_{\phi}(\mathbf{z})\|\pi(\mathbf{z}))$ alleviates this problem by adopting the aggregated posterior $q_{\phi}(\mathbf{z})$ for optimization instead of the encoder $q_{\phi}(\mathbf{z}|\mathbf{x})$. The authors of InfoVAE (Zhao et al., 2019) further provide the model family in which the KL term is replaced with other divergences. They introduce an alternative divergence $\mathcal{D}(q_{\phi}(\mathbf{z}),\pi(\mathbf{z}))$ and its weight $\lambda$ to conduct representation learning by the following training objective:

$$\underset{\boldsymbol{\theta},\boldsymbol{\phi}}{\mathrm{maximize}} \quad \mathbb{E}_{p_{\mathrm{data}}(\mathbf{x})}\left[\mathrm{ELBO}(\mathbf{x};\boldsymbol{\theta},\boldsymbol{\phi})\right] + I_{\phi}(\mathbf{x};\mathbf{z}) \tag{28}$$

$$= \mathbb{E}_{p_{\mathrm{data}}(\mathbf{x})}\mathbb{E}_{q_{\phi}(\mathbf{z}|\mathbf{x})}\left[p_{\boldsymbol{\theta}}(\mathbf{x}|\mathbf{z})\right] - \lambda\mathcal{D}(q_{\phi}(\mathbf{z}),\pi(\mathbf{z})). \tag{29}$$

In the original InfoVAE paper (Zhao et al., 2019), the authors reported that the Maximum-Mean Discrepancy (MMD) is the best choice for the divergence $\mathcal{D}$. The MMD divergence $\mathrm{MMD}(q_{\phi}(\mathbf{z}),\pi(\mathbf{z}))$ is defined as

$$\mathrm{MMD}(q_{\phi}(\mathbf{z}),\pi(\mathbf{z})) = \mathbb{E}_{q_{\phi}(\mathbf{z})}\mathbb{E}_{q_{\phi}(\mathbf{z}')}\left[k(\mathbf{z},\mathbf{z}')\right] + \mathbb{E}_{\pi(\mathbf{z})}\mathbb{E}_{\pi(\mathbf{z}')}\left[k(\mathbf{z},\mathbf{z}')\right]$$
$$- 2\mathbb{E}_{q_{\phi}(\mathbf{z})}\mathbb{E}_{\pi(\mathbf{z}')}\left[k(\mathbf{z},\mathbf{z}')\right], \tag{30}$$

where $k(\cdot,\cdot)$ is any universal kernel, such as the radial basis function kernel

$$k(\mathbf{z},\mathbf{z}') = \exp(-\|\mathbf{z} - \mathbf{z}'\|_2^2/\sigma^2) \tag{31}$$

for a constant $\sigma > 0$.

## A.2 VAE-BASED METHODS BASED ON HIERARCHICAL FACTORS

Several VAE-based methods postulate the existence of hierarchical factors as its meta-prior to learn representations with the abstractness of different levels (Sønderby et al., 2016; Zhao et al., 2017). These methods involve the change in their network architecture to utilize the feature hierarchy often captured in the hidden layers of deep neural networks.

### A.2.1 LADDER VARIATIONAL AUTOENCODER (LADDERVAE)

Ladder Variational Autoencoder (LadderVAE) (Sønderby et al., 2016) introduces hierarchical latent variables to the VAE model. Whereas the objective is still the ELBO, the LadderVAE model structure has hierarchical latent variables. The generative process is modeled as the Markov chain of several latent variable groups, and the inference model consists of deterministic feature encoders and the decoders shared with generative models. In the original paper (Sønderby et al., 2016), the authors claim that the LadderVAE models provide tighter log-likelihood lower bounds than the standard VAE.

### A.2.2 VARIATIONAL LADDER AUTOENCODER (VLADDERAE)

Variational Ladder Autoencoder (VLadderAE) (Zhao et al., 2017) is a VAE-based model for hierarchical factors. Instead of the hierarchical models based on Markov chains, the VLadderAE models introduce the hierarchical structure in the network architecture parameterizing the generative and the inference model. Since it constrains feature hierarchy by the process of feature extraction, VLadderAE also performs disentanglement, *e.g.*, the latent variables from different hidden convolutional layers capture textural or global features of visual data.

## A.3 VAE-BASED METHODS INVOLVING PRIOR LEARNING

The standard VAE model has a pre-defined prior, which may cause the discrepancy between the underlying data structure and the postulated prior (Dai & Wipf, 2019). Several methods overcome this problem by involving the prior itself in the training process.

### A.3.1 VAMPPRIOR

VampPrior (Tomczak & Welling, 2018) is a type of prior consisting of the mixture of the encoder distributions from several pseudo-input. The pseudo-inputs are introduced as trainable parameters, which are input into the encoder to build a mixture prior. Thus, the VAE models with VampPriors have trainable priors while retaining the main training procedure using the reparameterization trick to apply SGD.

### A.3.2 2-STAGE VAE

2-Stage VAE (Dai & Wipf, 2019) is a generative model with two probabilistic autoencoders. The process of 2-Stage VAE consists of two steps: (i) training a standard VAE using the given dataset as the input, and (ii) training another VAE using the latent variables of the previous VAE as the input. The 2-Stage VAE model attempts to overcome the discrepancy between the pre-defined prior and the learned latent representation by introducing the second VAE in stage (ii), which yields the prior training using the VAE in stage (i).

## A.4 WASSERSTEIN AUTOENCODER (WAE)

WAE (Tolstikhin et al., 2018) is a family of generative models whose autoencoder tries to estimate and minimize the primal form of the Wasserstein metric between the generative model $p_{\boldsymbol{\theta}}(\mathbf{x})$ and the data distribution $p_{\mathrm{data}}(\mathbf{x})$ using SGD with the following objective:

$$\underset{\boldsymbol{\theta}, \boldsymbol{\phi}}{\text{minimize}} \quad \mathbb{E}_{p_{\mathrm{data}}(\mathbf{x})} \mathbb{E}_{q_{\boldsymbol{\phi}}(\mathbf{z}|\mathbf{x})} \mathbb{E}_{p_{\boldsymbol{\theta}}(\mathbf{x}'|\mathbf{z})} \left[ d(\mathbf{x}, \mathbf{x}') \right] + \lambda \mathcal{D}(q_{\boldsymbol{\phi}}(\mathbf{z}), \pi(\mathbf{z})), \tag{32}$$

where $\lambda$ is a Lagrange multiplier, the generative model is defined as a latent variable model $p_{\boldsymbol{\theta}}(\mathbf{x}, \mathbf{z}) = \pi(\mathbf{z}) p_{\boldsymbol{\theta}}(\mathbf{x}|\mathbf{z})$ postulating the prior of the latent variables $\pi(\mathbf{z})$, and a conditional distribution $q_{\boldsymbol{\phi}}(\mathbf{z}|\mathbf{x})$ is a probabilistic encoder to optimize instead of all couplings supported on $\mathcal{X} \times \mathcal{X}$. The WAE

objective is indeed equivalent to that of InfoVAE (Zhao et al., 2019) in Eq. (27), which provides the OT-based perspective on VAE-based models. Following the InfoVAE (Zhao et al., 2019), we adopt the MMD for the divergence $\mathcal{D}$, which is denoted by "WAE-MMD" in the original WAE paper (Tolstikhin et al., 2018). Although the WAE-based approaches rewrite VAE-based objectives with the Wasserstein metric, these metrics are between $\mathbf{x}$-marginal distributions and do not directly include the latent space $\mathcal{Z}$. To learn representations $\mathbf{z}$, the Wasserstein-based objective is further modified (Gaujac et al., 2021).

### A.5  RELATIONAL REGULARIZED AUTOENCODER (RAE)

Relational Regularized Autoencoder (RAE) (Xu et al., 2020) is a variational autoencoding generative model with a regularization loss based on the fused Gromov-Wasserstein (FGW) metric. RAE introduces the FGW metric between the aggregated posterior and the latent prior as the regularization divergence to fortify the WAE constraint $\pi_{\boldsymbol{\theta}}(\mathbf{z}) = q_{\boldsymbol{\phi}}(\mathbf{z})$ introduced by Tolstikhin et al. (2018) for generative modeling. The FGW regularization is introduced with a weight hyperparameter $\beta \in [0, 1]$ and given as

$$\underset{\boldsymbol{\theta}, \boldsymbol{\phi}}{\text{minimize}} \quad \mathbb{E}_{p_{\text{data}}(\mathbf{x})} \mathbb{E}_{q_{\boldsymbol{\phi}}(\mathbf{z}|\mathbf{x})} \mathbb{E}_{p_{\boldsymbol{\theta}}(\mathbf{x}'|\mathbf{z})} [d(\mathbf{x}, \mathbf{x}')] + \lambda \mathcal{D}_{FGW}(q_{\boldsymbol{\phi}}(\mathbf{z}), \pi_{\boldsymbol{\theta}}(\mathbf{z}); \beta), \qquad (33)$$

where $\mathcal{D}_{FGW}$ denotes the FGW metric being the upper bound of the weighted sum of the Wasserstein and Gromov-Wasserstein metrics. The FGW metric $\mathcal{D}_{FGW}$ is given as

$$\mathcal{D}_{FGW}(q_{\boldsymbol{\phi}}(\mathbf{z}), \pi_{\boldsymbol{\theta}}(\mathbf{z}); \beta)$$

$$= \inf_{\gamma \in \mathcal{P}(q_{\boldsymbol{\phi}}(\mathbf{z}), \pi_{\boldsymbol{\theta}}(\mathbf{z}))} \left( (1-\beta) \mathbb{E}_{\gamma(\mathbf{z}, \mathbf{z}')} [d_{\mathcal{Z}}(\mathbf{z}, \mathbf{z}')] + \beta \mathbb{E}_{\gamma(\mathbf{z}_1, \mathbf{z}_1') \gamma(\mathbf{z}_2, \mathbf{z}_2')} [|d_{\mathcal{Z}}(\mathbf{z}_1, \mathbf{z}_2) - d_{\mathcal{Z}}(\mathbf{z}_1', \mathbf{z}_2')|] \right) \qquad (34)$$

$$\geq \quad (1-\beta) \underbrace{\inf_{\gamma \in \mathcal{P}(q_{\boldsymbol{\phi}}(\mathbf{z}), \pi_{\boldsymbol{\theta}}(\mathbf{z}))} \mathbb{E}_{\gamma(\mathbf{z}, \mathbf{z}')} [d_{\mathcal{Z}}(\mathbf{z}, \mathbf{z}')]}_{\text{Wasserstein term for direct comparison}}$$

$$+ \beta \underbrace{\inf_{\gamma \in \mathcal{P}(q_{\boldsymbol{\phi}}(\mathbf{z}), \pi_{\boldsymbol{\theta}}(\mathbf{z}))} \mathbb{E}_{\gamma(\mathbf{z}_1, \mathbf{z}_1') \gamma(\mathbf{z}_2, \mathbf{z}_2')} [|d_{\mathcal{Z}}(\mathbf{z}_1, \mathbf{z}_2) - d_{\mathcal{Z}}(\mathbf{z}_1', \mathbf{z}_2')|^2]}_{\text{Gromov-Wasserstein term for relational comparison}}, \qquad (35)$$

where $\mathcal{P}(q_{\boldsymbol{\phi}}(\mathbf{z}), \pi_{\boldsymbol{\theta}}(\mathbf{z}))$ is a set of all couplings whose marginals are $q_{\boldsymbol{\phi}}(\mathbf{z}), \pi_{\boldsymbol{\theta}}(\mathbf{z})$. The discrepancy between the prior $\pi_{\boldsymbol{\theta}}(\mathbf{z})$ and the aggregated posterior $q_{\boldsymbol{\phi}}(\mathbf{z})$ causes the degradation of generative performance since the processes of decoding $p_{\text{data}}(\mathbf{x}) q_{\boldsymbol{\phi}}(\mathbf{z}|\mathbf{x}) p_{\boldsymbol{\theta}}(\mathbf{x}|\mathbf{z})$ and generation $\pi_{\boldsymbol{\theta}}(\mathbf{z}) p_{\boldsymbol{\theta}}(\mathbf{x}|\mathbf{z})$ are modeled in different regions of the latent space. This formulation enables learning a prior distribution $q_{\boldsymbol{\phi}}(\mathbf{z})$ with flexibly assuming the structures of data, where the prior $\pi_{\boldsymbol{\theta}}(\mathbf{z})$ is modeled as a Gaussian mixture model the original settings by Xu et al. (2020). They aim at matching the distributions on the latent space $\mathcal{Z}$, which can have an identical dimensionality but may differ in terms of distance structure.

### A.6  IVI METHODS

Beyond the analytically tractable distributions, implicit distributions are applied to variational inference. An implicit distribution only requires its sampling method, which extends the variety of modeling and applications in variational inference and VAE-based models.

#### A.6.1  DENSITY RATIO ESTIMATION BY ADVERSARIAL DISCRIMINATORS

The density ratio estimation technique (Sugiyama et al., 2012) is essential to the mechanism of GANs (Goodfellow et al., 2014) and IVI methods (Huszár, 2017), which is conducted via an optimal discriminator $f^*$ between distributions $r(\mathbf{x})$ and $s(\mathbf{x})$ as

$$D_{\text{KL}}(r(\mathbf{x}) \| s(\mathbf{x})) = \mathbb{E}_{r(\mathbf{x})} \left[ \log \frac{r(\mathbf{x})}{s(\mathbf{x})} \right] = \mathbb{E}_{r(\mathbf{x})} \left[ \log \frac{f^*(\mathbf{x})}{1 - f^*(\mathbf{x})} \right]$$

$$= \mathbb{E}_{r(\mathbf{x})} [\log f^*(\mathbf{x}) - \log(1 - f^*(\mathbf{x}))], \qquad (36)$$

$$\text{where} \quad f^*(\mathbf{x}) = \arg \max_{f: \mathcal{X} \to (0,1)} \mathbb{E}_{r(\mathbf{x})} [\log f(\mathbf{x})] + \mathbb{E}_{s(\mathbf{x})} [\log(1 - f(\mathbf{x}))]. \qquad (37)$$

The discriminator is estimated via maximizing Eq. (37) with a neural network $f \approx f^*$. The training of discriminators often suffers from instability and mode collapse owing to its alternative parameter updates based on Eq. (36) and Eq. (37) (Arjovsky & Bottou, 2017; Arjovsky et al., 2017). One approach to tackle this problem is imposing the Lipschitz continuity on the discriminator based on the Kantorovich-Rubinstein duality (Arjovsky et al., 2017).

### A.6.2 ADVERSARIAL VARIATIONAL BAYES (AVB)

Adversarial Variational Bayes (AVB) (Mescheder et al., 2017) is an ELBO optimization method using the adversarial training process instead of the analytical KL term. Let us recall that the KL term in Eq. (1) is defined by the expected density ratio as

$$D_{\mathrm{KL}}(q_{\boldsymbol{\phi}}(\mathbf{z}|\mathbf{x})\|\pi(\mathbf{z})) = \mathbb{E}_{q_{\boldsymbol{\phi}}(\mathbf{z}|\mathbf{x})}\left[\frac{q_{\boldsymbol{\phi}}(\mathbf{z}|\mathbf{x})}{\pi(\mathbf{z})}\right]. \tag{38}$$

Adopting the density ratio trick (Sugiyama et al., 2012), the analytical KL term can be replaced with the optimal discriminator, which takes a data point $\mathbf{x}$ and its encoder sample $\mathbf{z} \sim q_{\boldsymbol{\phi}}(\mathbf{z}|\mathbf{x})$ to output the density ratio $q_{\boldsymbol{\phi}}(\mathbf{z}|\mathbf{x})/\pi(\mathbf{z})$. It enables implicit distributions in the prior while retaining the ELBO objective of variational inference.

### A.6.3 ADVERSARIALLY LEARNED INFERENCE (ALI) / BIDIRECTIONAL GENERATIVE ADVERSARIAL NETWORKS (BIGAN)

Adversarially Learned Inference (ALI) (Dumoulin et al., 2017) / Bidirectional Generative Adversarial Networks (BiGAN) (Donahue et al., 2017) are models introducing the distribution matching of the generative model and the inference model as implicit distributions. These models have been proposed in different papers (Dumoulin et al., 2017; Donahue et al., 2017); however, they share an equivalent methodology. One can draw samples from the generative model $\pi(\mathbf{z})p_{\boldsymbol{\theta}}(\mathbf{x}|\mathbf{z})$ by decoding prior samples and also from the inference model $p_{\mathrm{data}}(\mathbf{x})q_{\boldsymbol{\phi}}(\mathbf{z}|\mathbf{x})$ by encoding data points. Here the ALI/BiGAN models introduce a discriminator to estimate the Jensen-Shannon divergence between the generative model $p_{\boldsymbol{\theta}}(\mathbf{x}, \mathbf{z})$ and the inference model $q_{\boldsymbol{\phi}}(\mathbf{x}, \mathbf{z})$. The model matching between the encoder and the decoder also learns latent representations by the bidirectional mappings.

### A.6.4 VAE-GAN

VAE-GAN (Larsen et al., 2016) is a hybrid model based on VAE and GANs. The VAE-GAN models introduce a discriminator for the generative modeling w.r.t. the data $\mathbf{x}$ and utilize the hidden layers of the discriminator to model the decoder likelihood $p_{\boldsymbol{\theta}}(\mathbf{x}|\mathbf{z})$ along the manifolds supporting the data. It provides the outstanding performance of data generation to the VAE framework by measuring the similarity of data utilizing the GANs-like network architecture.

## B DETAILS OF PROPOSED METHOD

### B.1 MODELING DETAILS

The decoder $p_{\boldsymbol{\theta}}(\mathbf{x}|\mathbf{z})$ is modeled with a neural network $D_{\boldsymbol{\theta}} : \mathcal{Z} \to \mathbb{R}^M$ and its parameters $\boldsymbol{\theta}$ as

$$p_{\boldsymbol{\theta}}(\mathbf{x}|\mathbf{z}) = \delta(\mathbf{x} - D_{\boldsymbol{\theta}}(\mathbf{z})). \tag{39}$$

Following the standard VAE settings (Kingma & Welling, 2014), the encoder $q_{\boldsymbol{\phi}}(\mathbf{z}|\mathbf{x})$ is defined as a diagonal Gaussian parameterized by neural networks $\boldsymbol{\mu}_{\boldsymbol{\phi}} : \mathcal{Z} \to \mathbb{R}^M$ and $\boldsymbol{\sigma}_{\boldsymbol{\phi}}^2 : \mathcal{Z} \to \mathbb{R}_+^M$ with parameters $\boldsymbol{\phi}$ as

$$q_{\boldsymbol{\phi}}(\mathbf{z}|\mathbf{x}) = \mathcal{N}(\mathbf{z}|\boldsymbol{\mu}_{\boldsymbol{\phi}}(\mathbf{x}), \mathrm{diag}(\boldsymbol{\sigma}_{\boldsymbol{\phi}}^2(\mathbf{x}))). \tag{40}$$

For the distance functions $d_{\mathcal{X}}$ and $d_{\mathcal{Z}}$ in Eq. (7) and Eq. (9), we used the $L_2$ distance defined as

$$d_{\mathcal{X}}(\mathbf{x}, \mathbf{x}') = \frac{1}{\sqrt{2}}\|\mathbf{x} - \mathbf{x}'\|, \tag{41}$$

$$d_{\mathcal{Z}}(\mathbf{z}, \mathbf{z}') = \frac{1}{\sqrt{2}}\|\mathbf{z} - \mathbf{z}'\|. \tag{42}$$

As another choice, we also utilized the adversarially learned metric (Larsen et al., 2016) in Eq. (9). In the adversarially learned metric, the distance is measured in the feature space formed by the hidden outputs of the critic $f_\psi$. Let $h_\psi(\mathbf{x})$ denote the critic hidden outputs in which the critic takes $\mathbf{x}$ as its input. We can then define a distance $d'$ based on the adversarially learned metric as

$$d'(\mathbf{x}, \mathbf{x}') = \sqrt{d_\mathcal{X}(\mathbf{x}, \mathbf{x}')^2 + \frac{1}{2} \|h_\psi(\mathbf{x}) - h_\psi(\mathbf{x}')\|_2^2}. \tag{43}$$

Since the critic network $f_\psi(\mathbf{x}, \mathbf{z})$ has the Y-shaped architecture (see Appendix C.2) and concatenates $\mathbf{x}$-based features and $\mathbf{z}$-based features in one of the hidden layers to take a pair $(\mathbf{x}, \mathbf{z})$ as the inputs, we use the $\mathbf{x}$-side branch as $h_\psi(\mathbf{x})$.

### B.2  PRIOR DETAILS

**Neural Prior (NP).** Formally, the NP $\pi_\theta(\mathbf{z})$ with a neural network $g_\theta$ is defined as:

$$\pi_\theta(\mathbf{z}) = \int \pi(\boldsymbol{\epsilon}) \left| \det \frac{\partial g_\theta(\boldsymbol{\epsilon})}{\partial \boldsymbol{\epsilon}} \right| d\boldsymbol{\epsilon}, \tag{44}$$

$$\text{where} \quad \pi(\boldsymbol{\epsilon}) = \mathcal{N}(\boldsymbol{\epsilon}|\mathbf{0}, \mathbf{I}_L). \tag{45}$$

We can implement this class of prior with sampling noises $\boldsymbol{\epsilon}$ as $\mathbf{z} = g_\theta(\boldsymbol{\epsilon})$, avoiding the calculation of the integral.

**Factorized Neural Prior.** For disentanglement in the variational autoencoding settings, element-wise independence is often imposed on latent variables $\mathbf{z}$. Following the standard VAE settings (Kingma & Welling, 2014), we postulate $\mathcal{Z} = \mathbb{R}^L$, where the latent variables $\mathbf{z} \in \mathcal{Z}$ are expressed as an $L$-dimensional vector $\mathbf{z} = [z_1, z_2, \ldots, z_L]^\mathsf{T}$. As with the NP, the FNP class of prior is defined as

$$\pi_\theta(\mathbf{z}) = \prod_{i=1}^{L} \tilde{\pi}_\theta^{(i)}(z_i), \tag{46}$$

$$\text{where} \quad \tilde{\pi}_\theta^{(i)}(z_i) = \int \pi(\epsilon^{(i)}) \left| \frac{\partial \tilde{g}_\theta^{(i)}(\epsilon^{(i)})}{\partial \epsilon^{(i)}} \right| d\epsilon^{(i)}, \qquad (i = 1, 2, \ldots, L) \tag{47}$$

$$\pi(\epsilon^{(i)}) = \mathcal{N}(\epsilon^{(i)}|0, 1). \qquad (i = 1, 2, \ldots, L) \tag{48}$$

This prior can be implemented with $N$ disjoint neural networks, or 1-dimensional grouped convolutions. The difference between the NP and the FNP is element-wise independence, in which the prior $\pi_\theta(\mathbf{z})$ is factorized into distributions for each latent variable. Factorized priors enable disentanglement by obtaining a representation comprising independent factors of variation (Higgins et al., 2017a; Chen et al., 2018; Kim & Mnih, 2018).

### B.3  GRADIENT PENALTY

In the case of gradient penalty (Gulrajani et al., 2017), the maximization in Eq. (11) is further modified as

$$\underset{\psi}{\text{maximize}} \quad \mathcal{L}_D + \lambda_{GP} \mathbb{E}_{q_\phi(\mathbf{x}, \mathbf{z})} \mathbb{E}_{p_\theta(\mathbf{x}', \mathbf{z}')} \mathbb{E}_{\epsilon \sim \mathcal{U}(0,1)} \left[ \left( \|\nabla_{(\tilde{\mathbf{x}}, \tilde{\mathbf{z}})} f_\psi(\tilde{\mathbf{x}}, \tilde{\mathbf{z}})\|_2 - 1 \right)^2 \right], \tag{49}$$

where $\lambda_{GP} > 0$ is a constant, and $\tilde{\mathbf{x}} = \epsilon \mathbf{x} + (1 - \epsilon)\mathbf{x}'$ and $\tilde{\mathbf{z}} = \epsilon \mathbf{z} + (1 - \epsilon)\mathbf{z}'$ are interpolated samples by the random uniform noise $\epsilon$. We adopt $\lambda_{GP} = 10$ in all the experiments reported in this paper. Introducing the gradient penalty together with other techniques such as spectral normalization (Miyato et al., 2018) is effective and essential for adversarial learning in general (Chu et al., 2020; Miyato et al., 2018).

## C  EXPERIMENTAL DETAILS

For the reported experimental results, we used a single GPU of NVIDIA GeForce® RTX 2080 Ti, and a single run of the entire GWAE training process until convergence takes about eight hours.

## C.1 DATASET DETAILS

For the reported experiments in Section 4, we used the following datasets:

**MNIST (LeCun et al., 1998).** The MNIST dataset contains 70,000 handwritten digit images of 10 classes, comprising 60,000 training images and 10,000 test images. We used the original test set and randomly split the original training set into 54,000 training images and 6,000 validation images. We used the class information as its approximate factors of variation in the form of 10-dimensional dummy variables. This dataset is available online[2] in its original format or via the `torchvision` package[3] in the PyTorch (Paszke et al., 2019) tensor format. The MNIST dataset is licensed under the terms of the Creative Commons Attribution-Share Alike 3.0 license[4].

**CelebA (Liu et al., 2015).** The CelebA dataset contains 202,599 aligned face images with 40 binary attributes. We cropped $144 \times 144$ pixels in the center of the $178 \times 218$-sized aligned images in the original dataset to omit excessive backgrounds. We used the train/validation/test partitions that the original authors provided. We used the binary attributes as its approximate factors of variation in the form of 40-dimensional vectors. As in the website of this dataset[5], the CelebA dataset is available for non-commercial research purposes only.

**3D Shapes (Burgess & Kim, 2018).** The 3D Shapes dataset contains 480,000 synthetic images with six ground truth factors of variation. The images in this dataset contain a single-colored 3D object, a single-colored wall of a rectangular room, a single-colored floor. These images are procedurally generated from the independent factors of variation, *floor colour*, *wall colour*, *object colour*, *scale*, *shape*, and *orientation* (Burgess & Kim, 2018). We randomly split the entire dataset into 384,000/48,000/48,000 images for the train/validation/test set, respectively. Since the factor *shape* is a categorical variable in four classes, we converted it into four dummy variables to obtain quantitative factors of variation in the form of 9-dimensional vectors. The repository of this dataset[6] is licensed under Apache License 2.0[7].

**Omniglot (Lake et al., 2015).** The Omniglot dataset contains 1,623 images of hand-written characters from 50 different alphabets written by 20 different people. The images are $105 \times 105$-sized, binary-valued. We used this dataset as OoD samples over MNIST in the evaluations on the OoD detection utilizing cluster structure. The repository of this dataset[8] is licensed under the MIT License[9].

**CIFAR-10 (Krizhevsky & Hinton, 2009).** The CIFAR10 dataset contains 60,000 images with 10 classes, comprising 50,000 training images and 10,000 test images. The images are 32x32 color images in 10 natural image classes, such as airplane and cat. This dataset is provided online[10] without any specific license.

In all the datasets above, we used all the images as the raster (bitmap) representation and resized them to $64 \times 64$ pixels with three channels, where each image is a $3 \times 64 \times 64$-sized tensor value and $M = 12,288$. For gray-scale (one-channeled) images such as in MNIST, we repeated these images along the channel dimension three times to uniform these sizes to $3 \times 64 \times 64$ elements.

## C.2 ARCHITECTURE DETAILS

The architecture of neural networks in GWAE and the compared methods are built with convolutions and deconvolution (transposed convolution) in the same settings as shown in Tables 3 and 4. In all the experiments on GWAE, we applied the gradient penalty and the spectral normalization in

---

[2] http://yann.lecun.com/exdb/mnist/
[3] https://github.com/pytorch/vision
[4] https://creativecommons.org/licenses/by-sa/3.0/
[5] https://mmlab.ie.cuhk.edu.hk/projects/CelebA.html
[6] https://github.com/deepmind/3d-shapes
[7] http://www.apache.org/licenses/
[8] https://github.com/brendenlake/omniglot
[9] https://opensource.org/licenses/MIT
[10] https://www.cs.toronto.edu/~kriz/cifar.html

Table 3: Model architecture for the encoders in the GWAE models and the compared models. For the $64 \times 64$ RGB images used in the experiments, the input size is set to (Channels, Height, Width) = $(3, 64, 64)$. FC and Conv denote fully-connected (linear) layers and convolutional layers, respectively.

| Layer | Input Shape | Output Shape | Options |
|---|---|---|---|
| | | Inverse Sigmoid $\sigma^{-1}(x) = \log \frac{x}{1-x}$ | |
| Conv | $3 \times 64 \times 64$ | $32 \times 32 \times 32$ | kernel size=4, stride=2, padding=1 |
| | | SiLU activation (Hendrycks & Gimpel, 2016) | |
| Conv | $32 \times 32 \times 32$ | $64 \times 16 \times 16$ | kernel size=4, stride=2, padding=1 |
| | | SiLU activation (Hendrycks & Gimpel, 2016) | |
| Conv | $64 \times 16 \times 16$ | $128 \times 8 \times 8$ | kernel size=4, stride=2, padding=1 |
| | | SiLU activation (Hendrycks & Gimpel, 2016) | |
| Conv | $128 \times 8 \times 8$ | $256 \times 4 \times 4$ | kernel size=4, stride=2, padding=1 |
| | | SiLU activation (Hendrycks & Gimpel, 2016) | |
| FC | $256 \times 4 \times 4$ | $256$ | bias=True |
| | | SiLU activation (Hendrycks & Gimpel, 2016) | |
| FC | $256$ | $L$ for $\boldsymbol{\mu}$, $L$ for $\boldsymbol{\sigma}^2$ | bias=True |

the critic networks to impose the 1-Lipschitz continuity on the critic $f_\psi$, as shown in Table 7. In the neural samplers of GWAE models, we used the fully-connected architecture in Table 5 for NP and the grouped-convolutional architecture in Table 6 for FNP. We used fully connected layers for unconstrained priors in NP, and 1-dimensional grouped convolution layers (converting sequences with length 1 and $L$ channels) for factorized priors in FNP. For the optimizers of GWAE, we used RMSProp[11] with a learning rate of $10^{-4}$ for the main autoencoder network and used RMSProp with a learning rate of $5 \times 10^{-5}$ for the critic network. For all the compared methods except for GWAE, we used the Adam (Kingma & Ba, 2015) optimizer with a learning rate of $10^{-4}$. In the experiments, we used an equal batch size of 64 for all evaluated models. The batch size is relatively small, since the computational cost of GWAE for each batch is quadratic to the batch size $B$ and the GW estimation runs in time $O(NB)$ for each epoch using $\lceil N/B \rceil$ batches.

In the case that a batch normalization layer is introduced in the encoder outputs $q_\phi(\mathbf{z}_i|\mathbf{x}) = \mathcal{N}(\tilde{\boldsymbol{\mu}}(\mathbf{x}), \mathrm{diag}(\tilde{\boldsymbol{\sigma}}^2(\mathbf{x})))$, the mean and variance are computed w.r.t. the aggregated posterior $q_\phi(\mathbf{z})$ rather than the element-wise sample mean and variance of $L$-dimensional output values. The normalized parameters $(\tilde{\boldsymbol{\mu}}(\mathbf{x}), \tilde{\boldsymbol{\sigma}}(\mathbf{x}))$ against the original parameters $(\boldsymbol{\mu}(\mathbf{x}), \boldsymbol{\sigma}(\mathbf{x}))$ are given as

$$\tilde{\boldsymbol{\mu}}(\mathbf{x}) = \frac{\boldsymbol{\mu}(\mathbf{x}) - \mathbb{E}_{q_\phi(\mathbf{z})}[\mathbf{z}]}{\sqrt{\mathbb{V}_{q_\phi(\mathbf{z})}[\mathbf{z}]}}, \tag{50}$$

$$\tilde{\boldsymbol{\sigma}}^2(\mathbf{x}) = \frac{\boldsymbol{\sigma}^2(\mathbf{x})}{\mathbb{V}_{q_\phi(\mathbf{z})}[\mathbf{z}]}, \tag{51}$$

where the division is element-wise conducted, and $\mathbb{V}$ denotes the variance. The mean $\mathbb{E}_{q_\phi(\mathbf{z})}[\mathbf{z}]$ and variance $\mathbb{V}_{q_\phi(\mathbf{z})}[\mathbf{z}]$ are approximated using unbiased estimators consisting of mini-batch samples. Given a mini-batch index set $\mathcal{B} \subseteq \{1, 2, \dots, N\}$, the unbiased estimations are expressed using the law of total variance as

$$\mathbb{E}_{q_\phi(\mathbf{z})}[\mathbf{z}] \approx \frac{1}{\#\mathcal{B}} \sum_{i \in \mathcal{B}} \boldsymbol{\mu}(\mathbf{x}_i) =: \hat{\boldsymbol{\mu}}, \tag{52}$$

$$\mathbb{V}_{q_\phi(\mathbf{z})}[\mathbf{z}] \approx \frac{1}{\#\mathcal{B}} \sum_{i \in \mathcal{B}} \boldsymbol{\sigma}^2(\mathbf{x}_i) + \frac{1}{\#\mathcal{B} - 1} \sum_{i \in \mathcal{B}} (\boldsymbol{\mu}(\mathbf{x}) - \hat{\boldsymbol{\mu}})^2. \tag{53}$$

---

[11]https://www.cs.toronto.edu/~tijmen/csc321/slides/lecture_slides_lec6.pdf

Table 4: Model architecture for the decoders in the GWAE models and the compared models. The image shape is set to the same as Table 3. FC and DeConv denote fully-connected layers and deconvolutional layers, respectively.

| Layer | Input Shape | Output Shape | Options |
|-------|-------------|--------------|---------|
| FC | $L$ | 256 | bias=True |
| | SiLU activation (Hendrycks & Gimpel, 2016) | | |
| FC | 256 | $256 \times 4 \times 4$ | bias=True |
| | SiLU activation (Hendrycks & Gimpel, 2016) | | |
| DeConv | $256 \times 4 \times 4$ | $128 \times 8 \times 8$ | kernel size=4, stride=2, padding=1 |
| | SiLU activation (Hendrycks & Gimpel, 2016) | | |
| DeConv | $128 \times 8 \times 8$ | $64 \times 16 \times 16$ | kernel size=4, stride=2, padding=1 |
| | SiLU activation (Hendrycks & Gimpel, 2016) | | |
| DeConv | $64 \times 16 \times 16$ | $32 \times 32 \times 32$ | kernel size=4, stride=2, padding=1 |
| | SiLU activation (Hendrycks & Gimpel, 2016) | | |
| DeConv | $32 \times 32 \times 32$ | $3 \times 64 \times 64$ | kernel size=4, stride=2, padding=1 |
| | Sigmoid $\sigma(x) = \frac{1}{1+e^{-x}}$ | | |

Table 5: Model architecture for the samplers in the GWAE models with NP. FC denotes a fully-connected layer.

| Layer | Input Shape | Output Shape | Options |
|-------|-------------|--------------|---------|
| FC | $L$ | 256 | bias=True |
| SiLU activation (Hendrycks & Gimpel, 2016) | | | |
| FC | 256 | 256 | bias=True |
| SiLU activation (Hendrycks & Gimpel, 2016) | | | |
| FC | 256 | 256 | bias=True |
| SiLU activation (Hendrycks & Gimpel, 2016) | | | |
| FC | 256 | $L$ | bias=True |
| Batch Normalization with affine=False | | | |

Table 6: Model architecture for the samplers in the GWAE models with FNP. GroupConv denotes 1-dimensional grouped convolutional layers.

| Layer | Input Shape | Output Shape | Options |
|-------|-------------|--------------|---------|
| GroupConv | $L$ | 256 | bias=True, groups=$L$ |
| SiLU activation (Hendrycks & Gimpel, 2016) | | | |
| GroupConv | 256 | 256 | bias=True, groups=$L$ |
| SiLU activation (Hendrycks & Gimpel, 2016) | | | |
| GroupConv | 256 | 256 | bias=True, groups=$L$ |
| SiLU activation (Hendrycks & Gimpel, 2016) | | | |
| GroupConv | 256 | $L$ | bias=True, groups=$L$ |
| Batch Normalization with affine=False | | | |

## C.3 QUANTITATIVE EVALUATION DETAILS

For quantitative evaluations, we used the DCI scores (Eastwood & Williams, 2018) for disentanglement, the FID score (Heusel et al., 2017) for image generation, and the PSNR score for image reconstruction.

### C.3.1 DCI SCORES

The DCI scores (Eastwood & Williams, 2018) measure a representation in terms of disentangled representation learning. In the DCI scores, disentanglement is measured from three aspects: (i) each representation variable represents a single factor of variation, (ii) each factor of variation is expressed

Table 7: Model architecture for the critics in the GWAE models. We concatenated the outputs of the **x**-side and **z**-side branches and multiplied the concatenated outputs by $0.5$ to input into the stem network for the sake of the gradient norm, resulting in a Y-shaped network. We applied spectral normalization (Miyato et al., 2018) to all the layers in the critic networks and used the LeakyReLU (Maas et al., 2013) activation for the critic to retain the 1-Lipschitz continuity. FC and Conv denote fully-connected layers and convolutional layers, respectively.

| Layer | Input Shape | Output Shape | Options |
|---|---|---|---|
| | | **x**-side branch | |
| Conv | $3 \times 64 \times 64$ | $8 \times 32 \times 32$ | kernel size=4, stride=2, padding=1 |
| | LeakyReLU activation (Maas et al., 2013) with negative slope 0.2 | | |
| Conv | $8 \times 32 \times 32$ | $16 \times 16 \times 16$ | kernel size=4, stride=2, padding=1 |
| | LeakyReLU activation (Maas et al., 2013) with negative slope 0.2 | | |
| Conv | $16 \times 16 \times 16$ | $32 \times 8 \times 8$ | kernel size=4, stride=2, padding=1 |
| | LeakyReLU activation (Maas et al., 2013) with negative slope 0.2 | | |
| Conv | $32 \times 8 \times 8$ | $64 \times 4 \times 4$ | kernel size=4, stride=2, padding=1 |
| | LeakyReLU activation (Maas et al., 2013) with negative slope 0.2 | | |
| Conv | $64 \times 4 \times 4$ | $128 \times 2 \times 2$ | kernel size=4, stride=2, padding=1 |
| | LeakyReLU activation (Maas et al., 2013) with negative slope 0.2 | | |
| Conv | $128 \times 2 \times 2$ | $256 \times 1 \times 1$ | kernel size=4, stride=2, padding=1 |
| | LeakyReLU activation (Maas et al., 2013) with negative slope 0.2 | | |
| FC | 256 | 64 | bias=True |
| | | **z**-side branch | |
| FC | $L$ | 256 | bias=True |
| | LeakyReLU activation (Maas et al., 2013) with negative slope 0.2 | | |
| FC | 256 | 256 | bias=True |
| | LeakyReLU activation (Maas et al., 2013) with negative slope 0.2 | | |
| FC | 256 | 64 | bias=True |
| | LeakyReLU activation (Maas et al., 2013) with negative slope 0.2 | | |
| | | Stem network | |
| | | **z**-side branch | |
| FC | 64+64 | 256 | bias=True |
| | LeakyReLU activation (Maas et al., 2013) with negative slope 0.2 | | |
| FC | 256 | 256 | bias=True |
| | LeakyReLU activation (Maas et al., 2013) with negative slope 0.2 | | |
| FC | 256 | 1 | bias=True |
| | LeakyReLU activation (Maas et al., 2013) with negative slope 0.2 | | |

by a single representation variable, and (iii) a representation is informative w.r.t. the original data. The correspondence of variables and factors is computed via estimating the ground truth factors from the representation using random forest (Breiman, 2001). DCI Disentanglement (DCI-D) measures (i) the factor singleness for each variable. DCI Completeness (DCI-C) measures (ii) the variable singleness for each factor. DCI Informativeness (DCI-I) measures (iii) whether the representation is informative for estimating the ground truth factors. These metrics are computed via the variable importances (*e.g.*, the Gini impurity (Breiman, 2001)) of the random forest (Breiman, 2001), in which the random forest regressor estimates the ground truth factors using the representation variables. Using $L$-dimensional representation variables $\mathbf{z}$, $V$-dimensional factors $\mathbf{y}$ and their importance $R_{ij}$ of the $i$-th variable $z_i$

for the $k$-th factor $y_k$, the DCI-D and DCI-C scores for each variable and each factor are defined as

$$\text{DCI-D}_i = 1 + \sum_{k=1}^{V} p_{ik} \log_V p_{ik}, \qquad (i = 1, 2, \ldots, L) \qquad (54)$$

$$\text{where } p_{ik} = R_{ik} \bigg/ \sum_{j=1}^{V} R_{ij}, \qquad (55)$$

$$\text{DCI-C}_k = 1 + \sum_{i=1}^{L} q_{ik} \log_V q_{ik}, \qquad (k = 1, 2, \ldots, V) \qquad (56)$$

$$\text{where } q_{ik} = R_{ik} \bigg/ \sum_{j=1}^{V} R_{jk}. \qquad (57)$$

The DCI-D score for the entire variable set is given by the weighted sum $\sum_{i=1}^{L} \rho_i \text{DCI-D}_i$, where the weight $\rho_i$ is weighted importance $\rho_i = (\sum_{k=1}^{V} R_{ik})/(\sum_{i=1}^{L} \sum_{k=1}^{V} R_{ik})$. The DCI-C score for the entire factor set is given by the average score $1/V \sum_{k=1}^{V} \text{DCI-C}_k$. The DCI-D and DCI-C metrics take values within the range $[0, 1]$, where higher values indicate better performance. For DCI-I, we used the normalized definition by Zaidi et al. (2021) because the normalized DCI-I values are within the range $[0, 1]$ and the higher values mean better informativeness, while DCI-I score DCI-D$_{Original}$ is the estimation mean square error in the original definition. The DCI-I definition that we used is expressed as

$$\text{DCI-I} = 1 - 6 \times \text{DCI-I}_{Original}. \qquad (58)$$

Following the original paper (Eastwood & Williams, 2018), we set the number of random trees to 10 and decided the tree depth with cross-validation.

### C.3.2 FRÉCHET INCEPTION DISTANCE (FID)

Fréchet Inception Distance (FID) (Heusel et al., 2017) is a score for evaluating the quality of the generated images by generative models. The FID score is defined as the squared 2-Wasserstein metric between the features of the real images with mean $(\boldsymbol{\mu}_r, \boldsymbol{\Sigma}_r)$ and that of the generated images with mean $(\boldsymbol{\mu}_g, \boldsymbol{\Sigma}_g)$. Assuming that the features are normally distributed in the feature space, the FID score is expressed as

$$\text{FID} = W_2^2(\mathcal{N}(\boldsymbol{\mu}_r, \boldsymbol{\Sigma}_r), \mathcal{N}(\boldsymbol{\mu}_g, \boldsymbol{\Sigma}_g)) \qquad (59)$$

$$= \|\boldsymbol{\mu}_r - \boldsymbol{\mu}_g\|_2^2 + \text{tr}(\boldsymbol{\Sigma}_r + \boldsymbol{\Sigma}_g - 2(\boldsymbol{\Sigma}_r \boldsymbol{\Sigma}_g)^{\frac{1}{2}}). \qquad (60)$$

Since the Wasserstein metric measures the discrepancy between distributions, lower values indicate better generation performance in the FID score. Following the original FID paper (Heusel et al., 2017), we used the features obtained from the final pooling layer outputs of the Inception-v3 pre-trained in the ImageNet dataset (Deng et al., 2009).

### C.3.3 PEAK SIGNAL-TO-NOISE RATIO (PSNR)

For measuring the image reconstruction, we used the Peak Signal-to-Noise Ratio (PSNR) value. The PSNR value is defined as

$$\text{PSNR} = 20 \log_{10}(\text{MAX}) - 10 \log_{10}(\text{MSE}), \qquad (61)$$

where MAX denotes the maximum value of the pixel values, and MSE indicates the mean square error (MSE). In all the experiments conducted in Section 4, the value of MAX is set to MAX $= 1$ because the images input as a dataset data are scaled within the range $[0, 1]$.

### C.4 ISOMETRY COMPARISON

Regarding the evaluations in Section 4.2, we further conducted comparisons on isometry in Fig. 4. The results show that the GWAE models provide more isometric autoencoders compared with other VAE-based representation learning methods. The existing VAE-based methods did not yield as far as GWAE, which supports that the GW metric works as a different objective class from the ELBO. This implies that the GW metric loss substantially affects the training procedure of learning representations.

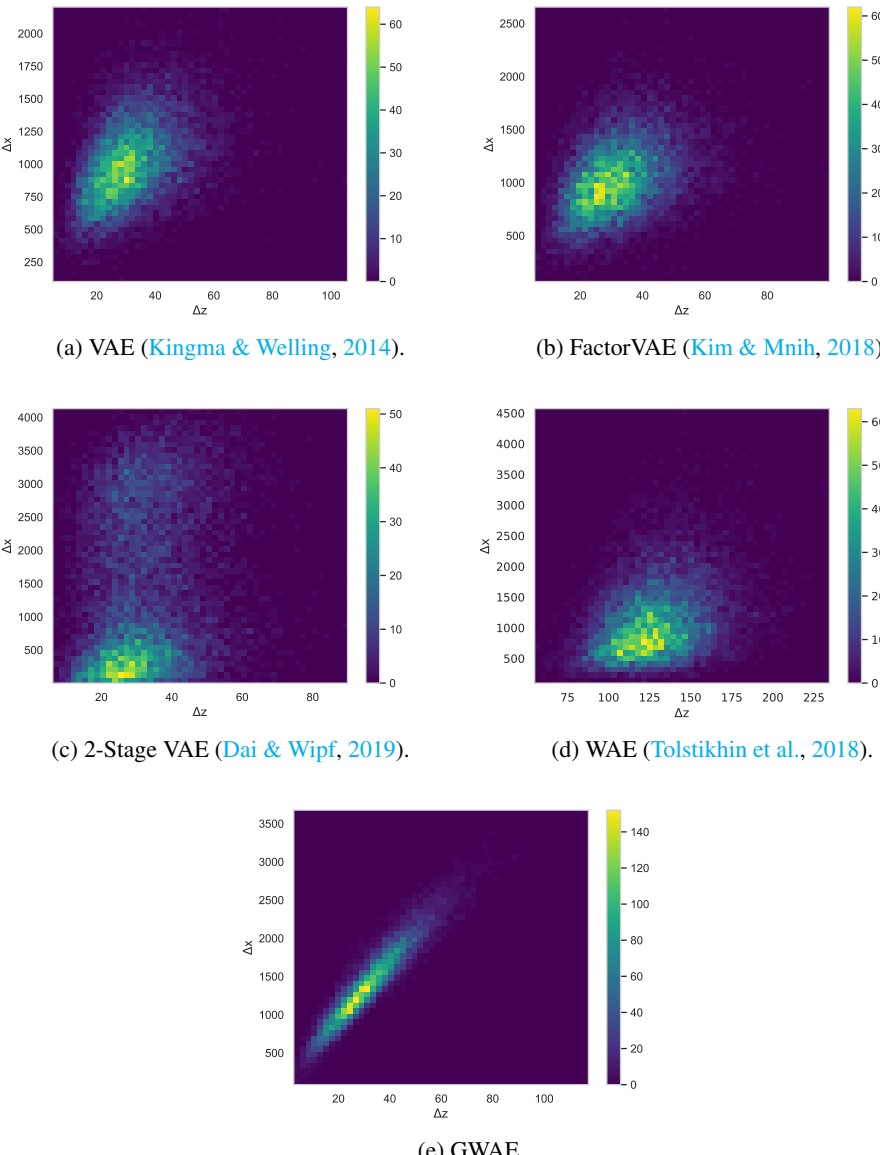

(a) VAE (Kingma & Welling, 2014).

(b) FactorVAE (Kim & Mnih, 2018).

(c) 2-Stage VAE (Dai & Wipf, 2019).

(d) WAE (Tolstikhin et al., 2018).

(e) GWAE.

Figure 4: Histograms of the differences in MNIST (LeCun et al., 1998). Each histogram consists of 10,000 samples of $(\Delta x, \Delta z)$, where $\Delta x$ (vertical) and $\Delta z$ (horizontal) respectively denote the differences $\Delta x = d_{\mathcal{X}}(\mathbf{x}, \mathbf{x}')$ and $\Delta z = d_{\mathcal{Z}}(\mathbf{z}, \mathbf{z}')$ of two generative samples $(\mathbf{x}, \mathbf{z}), (\mathbf{x}', \mathbf{z}') \sim p_{\boldsymbol{\theta}}(\mathbf{x}, \mathbf{z})$. In all reported results including FactorVAE (Kim & Mnih, 2018) ($\gamma$=3), WAE (Tolstikhin et al., 2018), and GWAE (NP, $\lambda_D$=1, $\lambda_W$=1, $\lambda_{\mathcal{H}}$=1), the latent dimension $L$ was set to $L = 16$, and their priors were set to the standard Gaussian.

Table 8: The effect of prior family selection in the GWAE model. The same settings in Table 1 are applied to all the reported models.

| Model | DCI-C ↑ | DCI-D ↑ | DCI-I ↑ |
|---|---|---|---|
| GWAE (NP) | 0.3966 | 0.3113 | 0.9403 |
| GWAE (FNP) | **0.9080** | **0.7024** | **0.9966** |
| GWAE (GMP) | 0.4247 | 0.4373 | 0.9655 |

## C.5 TRAINING PROCESS

We present the training process of GWAE in Fig. 5. Although the objective seems complex for its composition of four different losses, the training process successfully converged and the values of the terms $\mathcal{L}_{GW}$, $\mathcal{L}_W$, and $\mathcal{L}_D$ jointly descended in the most part of training. Although the term $-\mathcal{R}_\mathcal{H}$ increased, its values did not diverge to prevent the degenerate solutions. These results imply that the three different losses $\mathcal{L}_{GW}$, $\mathcal{L}_W$, and $\mathcal{L}_D$ did not conflict during the training process even for the complicated data, balancing these three terms against $-\mathcal{R}_\mathcal{H}$ as in the trade-off of the reconstruction against the regularization in $\beta$-VAE (Higgins et al., 2017a; Tschannen et al., 2018).

## C.6 PRIOR FAMILY SELECTION

We show the effect of prior family selection regarding a meta-prior, disentanglement, in Table 8. While GWAE models with the NP and GMP retain the informativeness of the FNP, the other two priors than FNP did not comparably disentangle the latent factors. Although the NP covers a more general family of prior, these results suggest that choosing a prior family suitable to the postulated meta-prior greatly facilitates learning representations.

## C.7 QUALITATIVE EVALUATIONS OF GENERATION AND RECONSTRUCTION

We show the reconstructed images by GWAE and state-of-the-art variational autoencoding methods in Fig. 6. The shown images are the first ten samples of the test split in the CelebA (Liu et al., 2015) dataset under the latent size $L = 64$. Compared with the other methods, the reconstruction of the GWAE model tends to retain edges (see the bottom rows of Fig. 6), while VAE-based models generate smooth, blurry images due to the noise injected in the latent space to perform probabilistic modeling and manifold learning. We also show the reconstruction results of MNIST (LeCun et al., 1998) in Fig. 7 and CIFAR-10 (Krizhevsky & Hinton, 2009) in Fig. 8. These results support that the GWAE models consistently perform autoencoding also in a more simple dataset (MNIST). In a more complex dataset (CIFAR-10), the GWAE model attained the best evaluation in generation albeit its reconstruction, suggesting that the GWAE model successfully captured the abstract structure of data rather than reconstructed the given images. This difference highlights the difference in their objectives, *i.e.*, the GW objective aims at distribution matching in the latent space, while the $\beta$-VAE (Higgins et al., 2017a) objective with $\beta < 1$ puts weight on reconstruction.

We further study the generated images by GWAE and state-of-the-art VAE-based generative models in Fig. 9. These qualitative results show that the GWAE generation successfully obtains a diverse set of images compared with those of state-of-the-art autoencoding generative models. The ALI model (Dumoulin et al., 2017) (Fig. 9 (a)) also generates various images by the distribution matching of bidirectional models, but the generated images have wavy contours, failing at composing images with a consistent appearance owing to the lack of an autoencoding process. Although the VAE-GAN model (Larsen et al., 2016) (Fig. 9 (b)) adequately yields organized images with smooth textures, the azimuth of these images is less diverse, *i.e.*, the great majority of the images are facing forward or looking slightly sideways. The images generated by 2-Stage VAE (Dai & Wipf, 2019) (Fig. 9 (c)) have diverse azimuth, color, and background; however, these images tend to incline toward the majority attributes, *e.g.*, not wearing eyeglasses or sunglasses. The GWAE model (Fig. 9 (d)) successfully generates facial images with various skin colors, diversified backgrounds, and assorted facial expressions (*e.g.*, wearing a mustache). These results imply that the GWAE models also function as generative models while it has been built as a representation learning method owing to the

Table 9: The ablation study on generation and reconstruction in CelebA (Liu et al., 2015). The same settings as Table 2 are applied in these experiments.

| Model | FID (Heusel et al., 2017) ↓ | PSNR [dB] ↑ |
|---|---|---|
| GWAE (NP) | **45.3** | **22.82** |
| GWAE (NP) w/o $\mathcal{L}_W$ | 233.7 | 9.80 |
| GWAE (NP) w/o $\mathcal{L}_D$ | 403.8 | 18.63 |
| GWAE (NP) w/ MMD $\mathcal{L}_D$ | 158.4 | 22.61 |
| GWAE (NP) w/ $\mathcal{Z}$-only critic | 102.4 | 22.41 |
| GWAE (NP) w/o $\mathcal{R}_{\mathcal{H}}$ | 179.6 | 21.57 |
| GWAE (NP, $\rho = \xi$) | 123.5 | 16.03 |

collateral condition $p_{\boldsymbol{\theta}}(\mathbf{x}, \mathbf{z}) \approx q_{\boldsymbol{\phi}}(\mathbf{x}, \mathbf{z})$ in Eq. (11) and the generative modeling $p_{\boldsymbol{\theta}}(\mathbf{x}) \approx p_{\text{data}}(\mathbf{x})$ as its necessary condition.

## C.8 ABLATION STUDY

We conducted the ablation study of the losses and regularizations introduced in Eq. (13). Table 9 shows the results of the ablation study of the three sub-constraints $\mathcal{L}_W$, $\mathcal{L}_D$, and $\mathcal{R}_{\mathcal{H}}$. The ablations yielded the performance degradation of GWAE, especially in $\mathcal{L}_W$. These results suggest the necessity of each regularization term and reveal their roles in representation learning.

**Ablation of $\mathcal{L}_W$.** The ablation of the term $\mathcal{L}_W$ brought low-quality reconstruction, which suggests that $\mathcal{L}_W$ works as the autoencoding constraint as can be seen from taking the reconstruction loss in $\mathcal{L}_W$. It also reduced generation capability as well as reconstruction, suggesting that the generative modeling via autoencoding is inherited from the variational autoencoding architecture of VAEs (Kingma & Welling, 2014).

**Ablation of $\mathcal{L}_D$.** Without the term $\mathcal{L}_D$, the GWAE models suffer from the lack of distribution matching in data generation, while it successfully conducted data reconstruction. These phenomena could be caused by the discrepancy between the encoded latent distribution $q_{\boldsymbol{\phi}}(\mathbf{z})$ and the prior $\pi_{\boldsymbol{\theta}}(\mathbf{z})$. Similar results are also obtained in the ablation of the merged sufficient condition (see Eq. (11)) for the regularization $\mathcal{L}_D$, where $\mathcal{L}_D$ is defined as the MMD loss between the prior $\pi_{\boldsymbol{\theta}}(\mathbf{z})$ and the encoded latent $q_{\boldsymbol{\phi}}(\mathbf{z})$, as in the WAE-MMD model (Tolstikhin et al., 2018). This choice of $\mathcal{L}_D$ on the low-dimensional space $\mathcal{Z}$ appears to be a replacement for the Kantorovich potential adversarially learned in the high-dimensional joint space $\mathcal{X} \times \mathcal{Z}$; however, lacking the merged sufficient condition seems to have caused the crucial reduction of generation performance as in the gross ablation of $\mathcal{L}_D$. These results imply that the term $\mathcal{L}_D$ with adversarial learning regularizes the generative model $p_{\boldsymbol{\theta}}(\mathbf{x}, \mathbf{z})$ to match the inference $q_{\boldsymbol{\phi}}(\mathbf{x}, \mathbf{z})$.

**Ablation of $\mathcal{R}_{\mathcal{H}}$.** Removing $\mathcal{R}_{\mathcal{H}}$ slightly increased the reconstruction error but deteriorated the generation quality. To confirm this behavior, we also show the samples generated by the GWAE model without the regularization $\mathcal{R}_{\mathcal{H}}$ in Fig. 10 and its reconstruction in Fig. 11. These qualitative results that the decoder without $\mathcal{R}_{\mathcal{H}}$ successfully reconstructs the images from the inference $q_{\boldsymbol{\phi}}(\mathbf{z})$ but generates corrupted images from the prior $\pi_{\boldsymbol{\theta}}(\mathbf{z})$. It suggests the "hole" problem (Rezende & Viola, 2018b) in the degenerate solution, where each data point is mapped at a single latent point to cover the zero-measure area of the latent space and the latent space is almost everywhere not covered by the inference $q_{\boldsymbol{\phi}}(\mathbf{z})$. Thus, the entropy regularization $\mathcal{R}_{\mathcal{H}}$ seems to have worked for retaining the probabilistic mappings in the encoder $q_{\boldsymbol{\phi}}(\mathbf{z}|\mathbf{x})$ to avoid this phenomenon.

In addition, for ablating $\rho = 1$, we also experimented with the $\rho = \xi$ settings that appear to be intuitively natural although causing an unstable training process due to the outlier samples in $\mathcal{L}_{GW}$. The GWAE model with $\rho = \xi$ suffered from performance degradation both in the generation and reconstruction, suggesting that our settings $\rho = 1 \leq \xi$ affect the learning process of the entire model.

## C.9 THE META-PRIOR EFFECT ON GW MINIMIZATION AND ESTIMATION

For a further inspection of Section 4.2, we also studied the GW minimization and estimation using FNP in Fig. 12. Compared with the NP case in Fig. 1, GWAE with FNP presents less stable and

more biased estimation and minimization. The learning curve of $\mathcal{L}_{GW}$ in Fig. 12a is largely biased in the first 40 epochs and then seems to be converged at approximately 3.2, a higher value than that of Fig. 1a (lower than 2). The isometry histogram also suggests the degradation of GW minimization in FNP. In Fig. 12b, more samples fell in off-diagonal areas, showing that the isometry is less tight than that of Fig. 1b. These results are presumably due to the mismatch of disentanglement meta-prior in MNIST (LeCun et al., 1998) because one of the major generative factors of MNIST images is the kind of digits, a categorical variable typically learned as one-hot variables in contrast to the factorization imposed by FNP.

### C.10   PRIORS IN CLUSTERING STRUCTURE

For more detailed investigation of the capture of clustering structure studied in Fig. 3, we further study the latent spaces of VAE (Kingma & Welling, 2014), DAGMM (Zong et al., 2018), and GWAE with GMP. The t-SNE visualization (van der Maaten & Hinton, 2008) of the latent spaces are shown in Fig. 13, which suggests that the GWAE model with GMP clearly captured the clustering structure in its latent space. The prior of VAE (Kingma & Welling, 2014) is defined as the standard Gaussian $\mathcal{N}(\mathbf{0}, \mathbf{I}_L)$ which does not consist of multiple clusters. The learned prior of DAGMM contains multiple clusters; however, adjacent clusters were overlapping to some extent. From the learned prior in GWAE, we can observe clear clusters densely concentrating themselves and separating each other. These results support the quantitative OoD results in Fig. 3, in which the GWAE model outperforms the other two models with and without explicit clustering modeling, respectively.

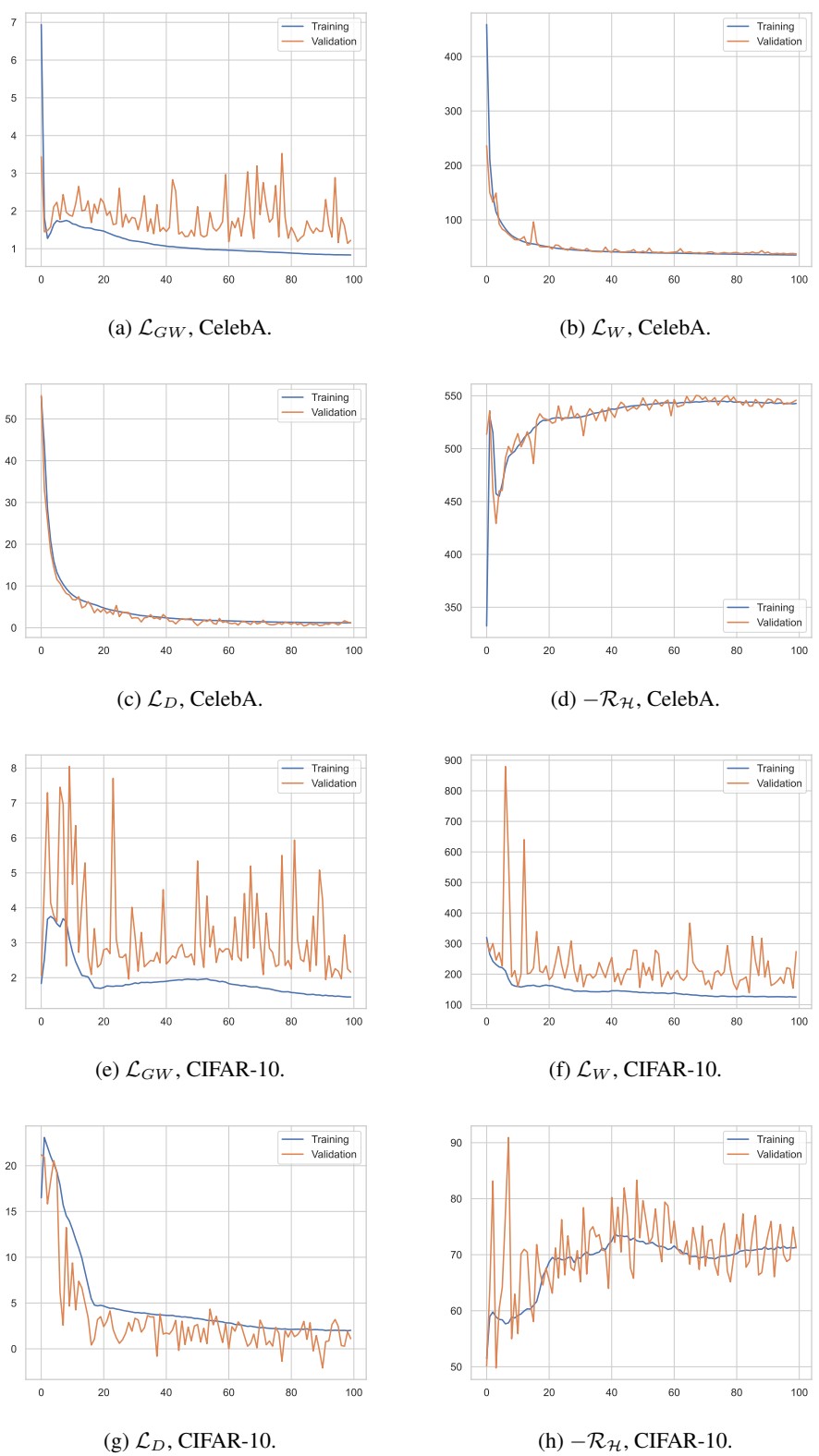

Figure 5: The training process of a GWAE model. The model is trained using NP and $\lambda_{\mathcal{H}} = \lambda_W = \lambda_{\mathcal{H}} = 1$. The plot (a)–(d) are training curves during one trial of training using CelebA (Liu et al., 2015), and (e)–(h) are using CIFAR-10 (Krizhevsky & Hinton, 2009). In each plot, the horizontal axis represents the number of epochs elapsed, and the vertical axis expresses the loss value. The blue and orange curves represent the training and validation losses, respectively.

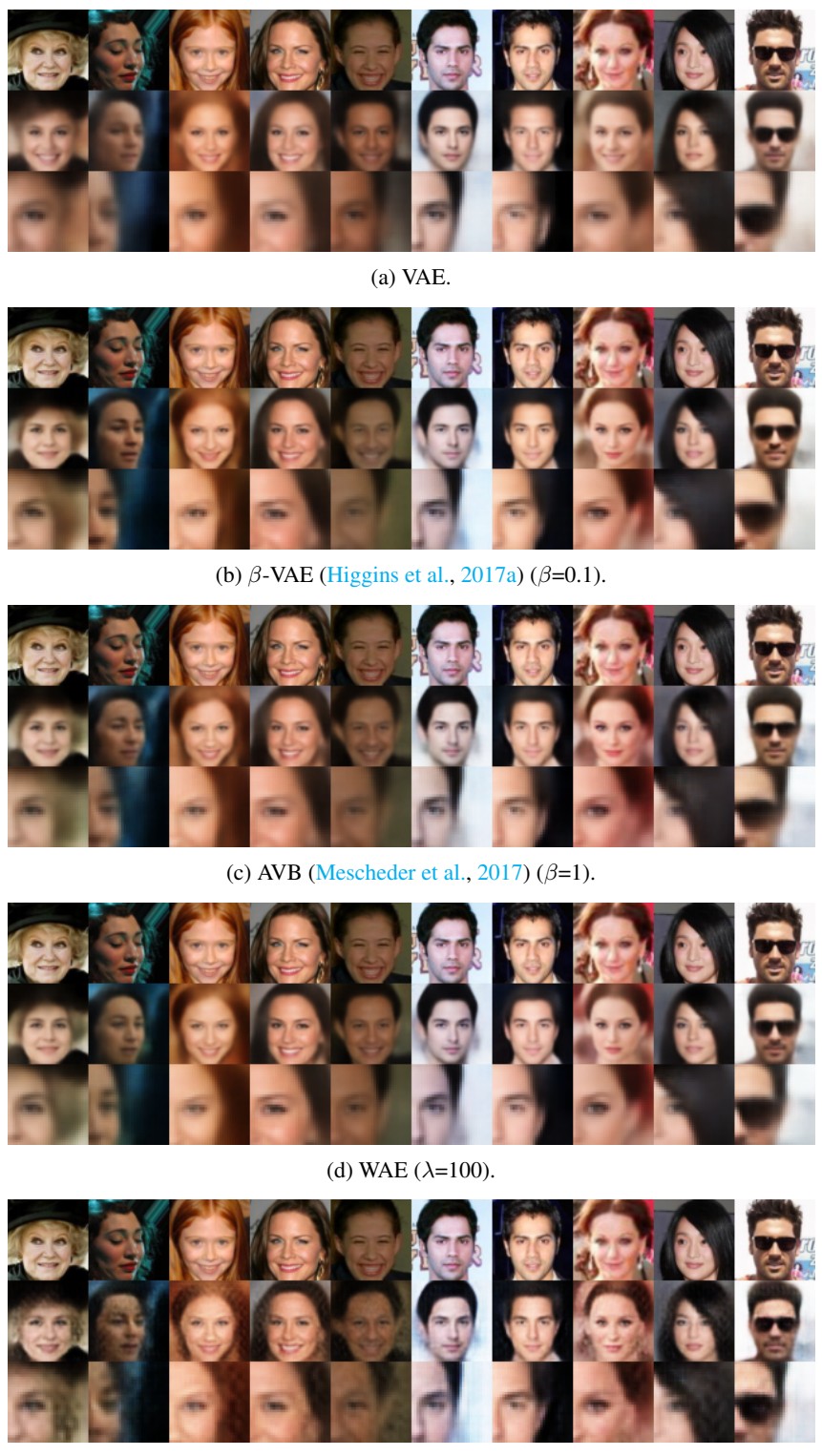

(a) VAE.

(b) $\beta$-VAE (Higgins et al., 2017a) ($\beta$=0.1).

(c) AVB (Mescheder et al., 2017) ($\beta$=1).

(d) WAE ($\lambda$=100).

(e) GWAE (NP, $\lambda_D$=1, $\lambda_W$=10, $\lambda_\mathcal{H}$=0.0001).

Figure 6: Reconstructed images in CelebA (Liu et al., 2015). The images denote original data samples (top rows), reconstructed images (middle rows), and zoomed reconstructions (bottom rows). Each column corresponds to one data instance in the test set.

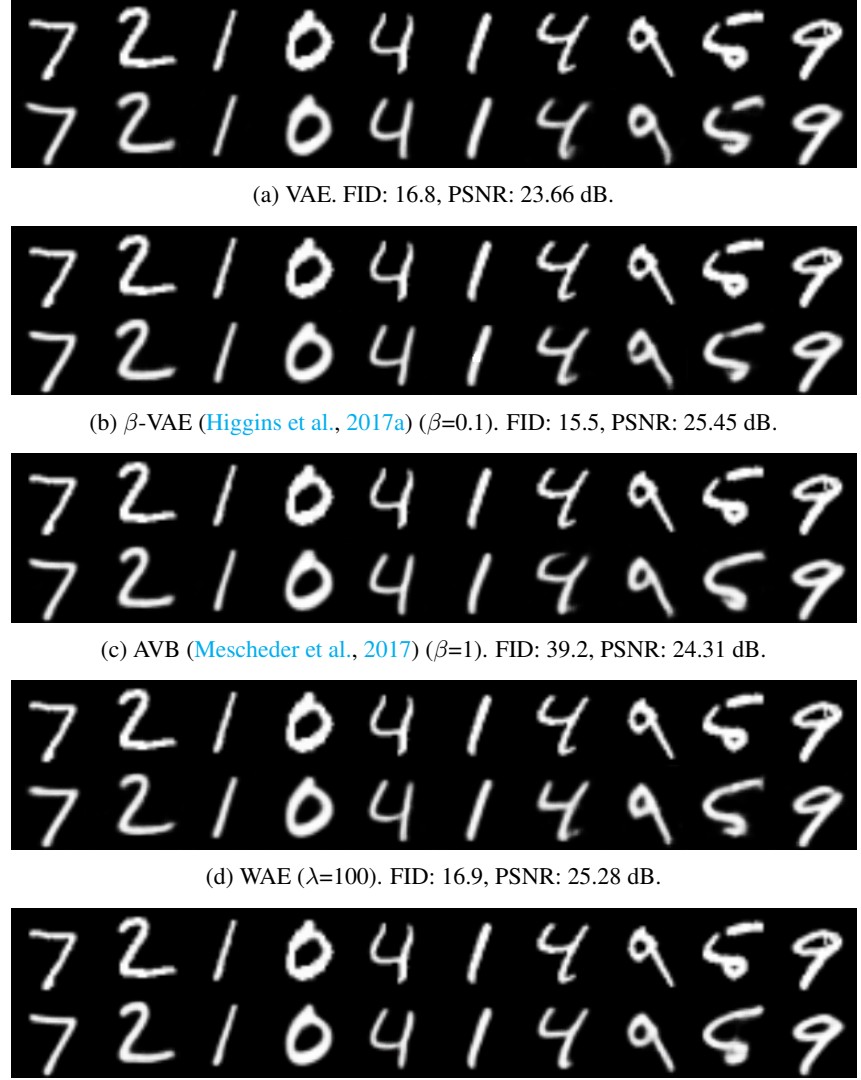

(a) VAE. FID: 16.8, PSNR: 23.66 dB.

(b) $\beta$-VAE (Higgins et al., 2017a) ($\beta$=0.1). FID: 15.5, PSNR: 25.45 dB.

(c) AVB (Mescheder et al., 2017) ($\beta$=1). FID: 39.2, PSNR: 24.31 dB.

(d) WAE ($\lambda$=100). FID: 16.9, PSNR: 25.28 dB.

(e) GWAE (NP, $\lambda_D$=1, $\lambda_W$=10, $\lambda_{\mathcal{H}}$=0.0001). FID: 14.4, PSNR: 26.11 dB.

Figure 7: Reconstructed images in MNIST (LeCun et al., 1998). The images denote original data samples (top rows), reconstructed images (bottom rows). Each column corresponds to one data instance in the test set.

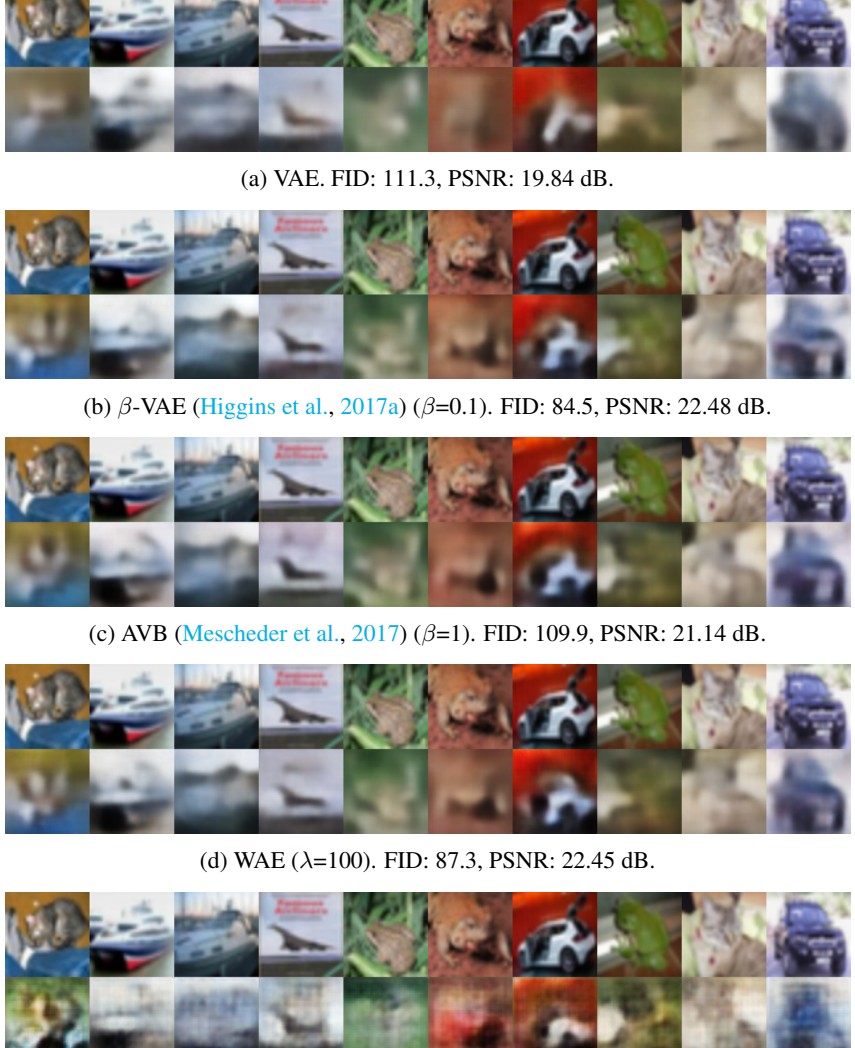

(a) VAE. FID: 111.3, PSNR: 19.84 dB.

(b) $\beta$-VAE (Higgins et al., 2017a) ($\beta$=0.1). FID: 84.5, PSNR: 22.48 dB.

(c) AVB (Mescheder et al., 2017) ($\beta$=1). FID: 109.9, PSNR: 21.14 dB.

(d) WAE ($\lambda$=100). FID: 87.3, PSNR: 22.45 dB.

(e) GWAE (NP, $\lambda_D$=1, $\lambda_W$=10, $\lambda_{\mathcal{H}}$=0.0001). FID: 59.9, PSNR: 17.64 dB.

Figure 8: Reconstructed images in CIFAR-10 (Krizhevsky & Hinton, 2009). The images denote original data samples (top rows), reconstructed images (bottom rows). Each column corresponds to one data instance in the test set.

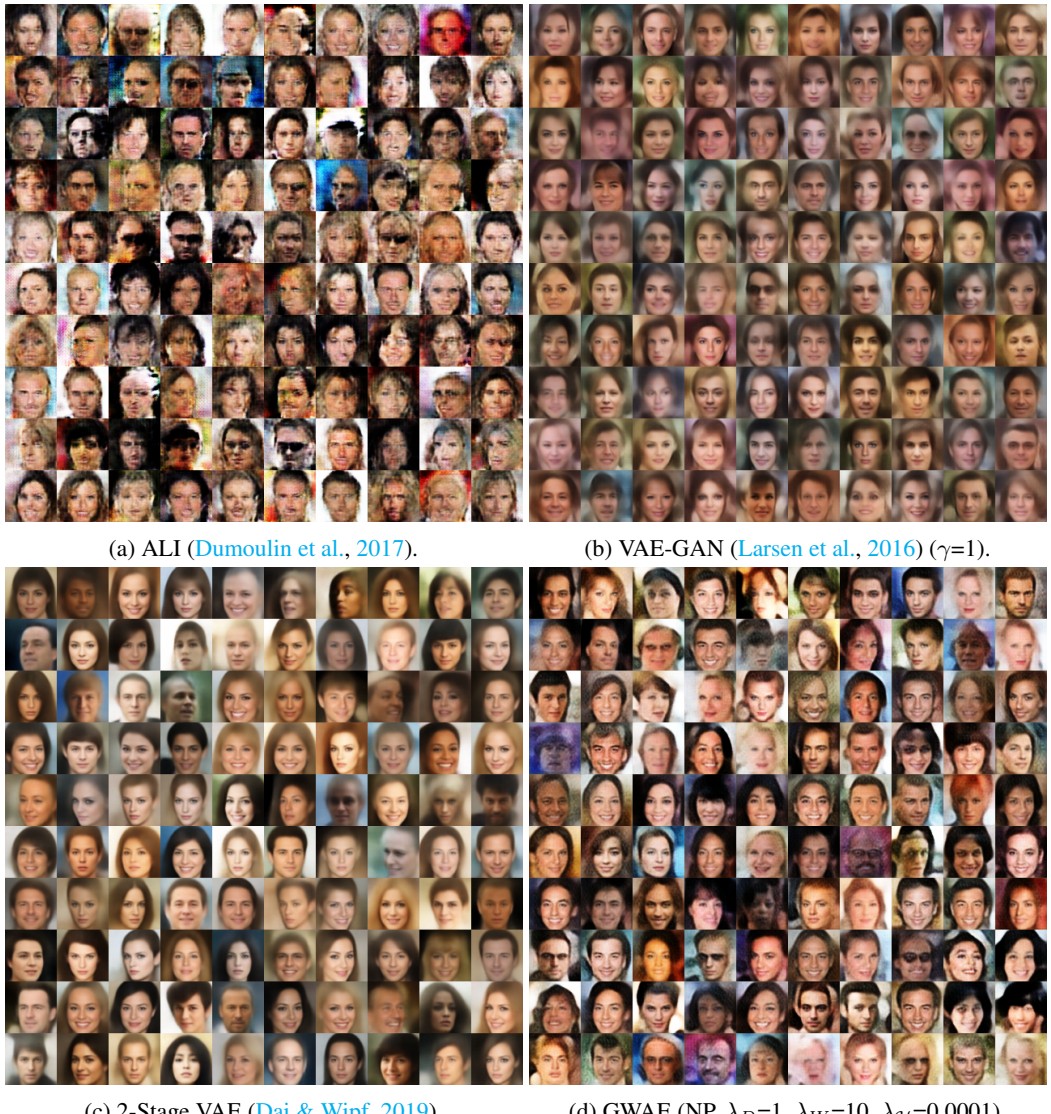

(a) ALI (Dumoulin et al., 2017).

(b) VAE-GAN (Larsen et al., 2016) ($\gamma$=1).

(c) 2-Stage VAE (Dai & Wipf, 2019).

(d) GWAE (NP, $\lambda_D$=1, $\lambda_W$=10, $\lambda_{\mathcal{H}}$=0.0001).

Figure 9: Generated images in CelebA (Liu et al., 2015). We show 100 images sampled from the generative model $p_{\boldsymbol{\theta}}(\mathbf{x})$ without conducting cherry-picking.

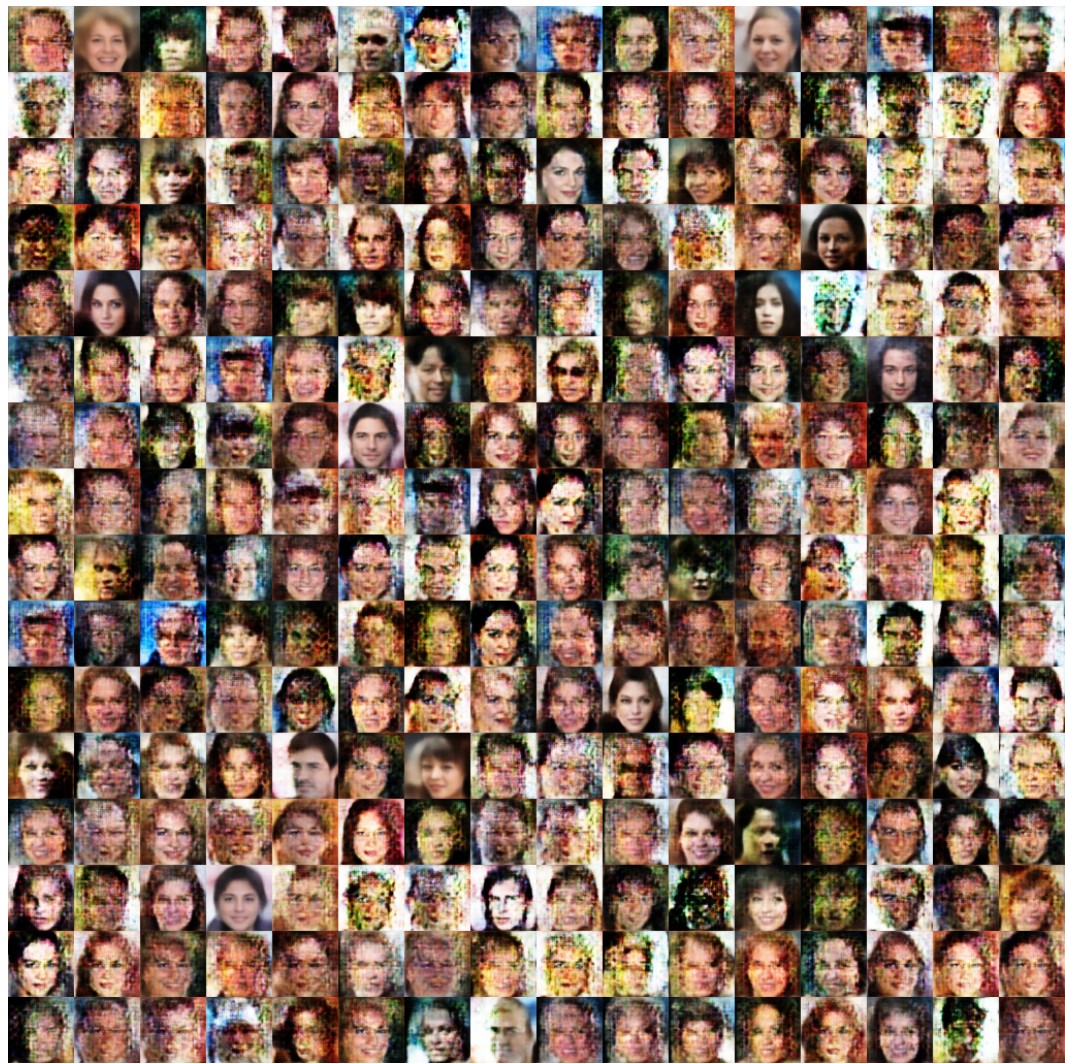

Figure 10: Generated images in CelebA (Liu et al., 2015) using the GWAE model without the regularization term $\mathcal{R}_{\mathcal{H}}$.

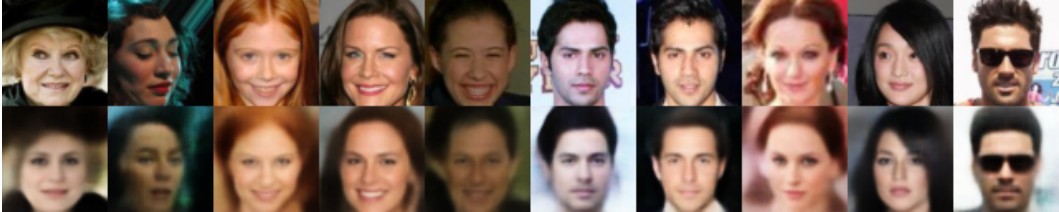

Figure 11: Reconstructed images in CelebA (Liu et al., 2015) using the GWAE model without the regularization term $\mathcal{R}_{\mathcal{H}}$. Each column corresponds to one test data instance. The rows denote original (top) and reconstructed (bottom) images.

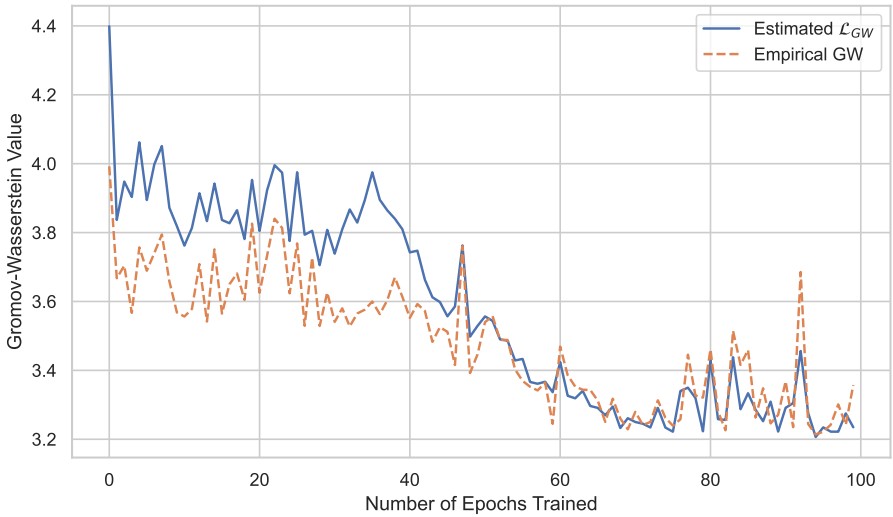

(a) The estimation of the GW metric using FNP.

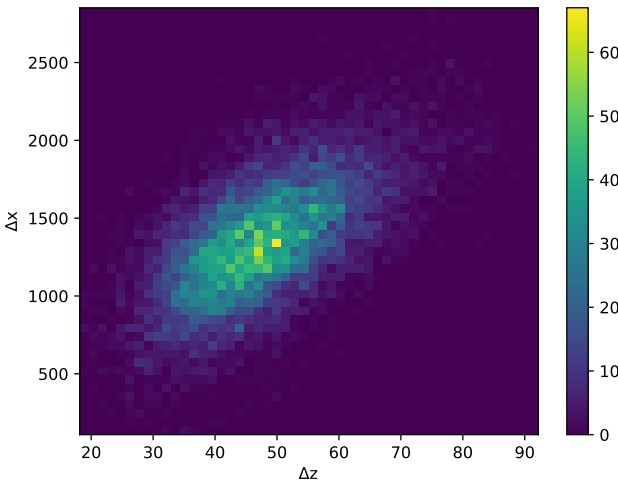

(b) The isometry in GWAE with FNP.

Figure 12: The estimation and minimization of the GW metric. This trial of training is conduced in GWAE (**FNP**, $\lambda_D$=1, $\lambda_W$=1, $\lambda_{\mathcal{H}}$=1) using the MNIST (LeCun et al., 1998) dataset, which is the same settings as Fig. 1 except for FNP. (a) The curves show the GW values estimated by the loss term $\mathcal{L}_{GW}$ (solid, blue) and the empirical GW computed by the POT package (Flamary et al., 2021) (dashed, orange). The values are computed using the validation set. (b) The axes $\Delta x = d_{\mathcal{X}}(\mathbf{x}, \mathbf{x}')$ (vertical) and $\Delta z = d_{\mathcal{Z}}(\mathbf{z}, \mathbf{z}')$ (horizontal) respectively denote the difference in the data and latent spaces between generated samples $(\mathbf{x}, \mathbf{z}), (\mathbf{x}', \mathbf{z}') \sim p_{\boldsymbol{\theta}}(\mathbf{x}, \mathbf{z})$. The histogram contains 10,000 generated sample pairs.

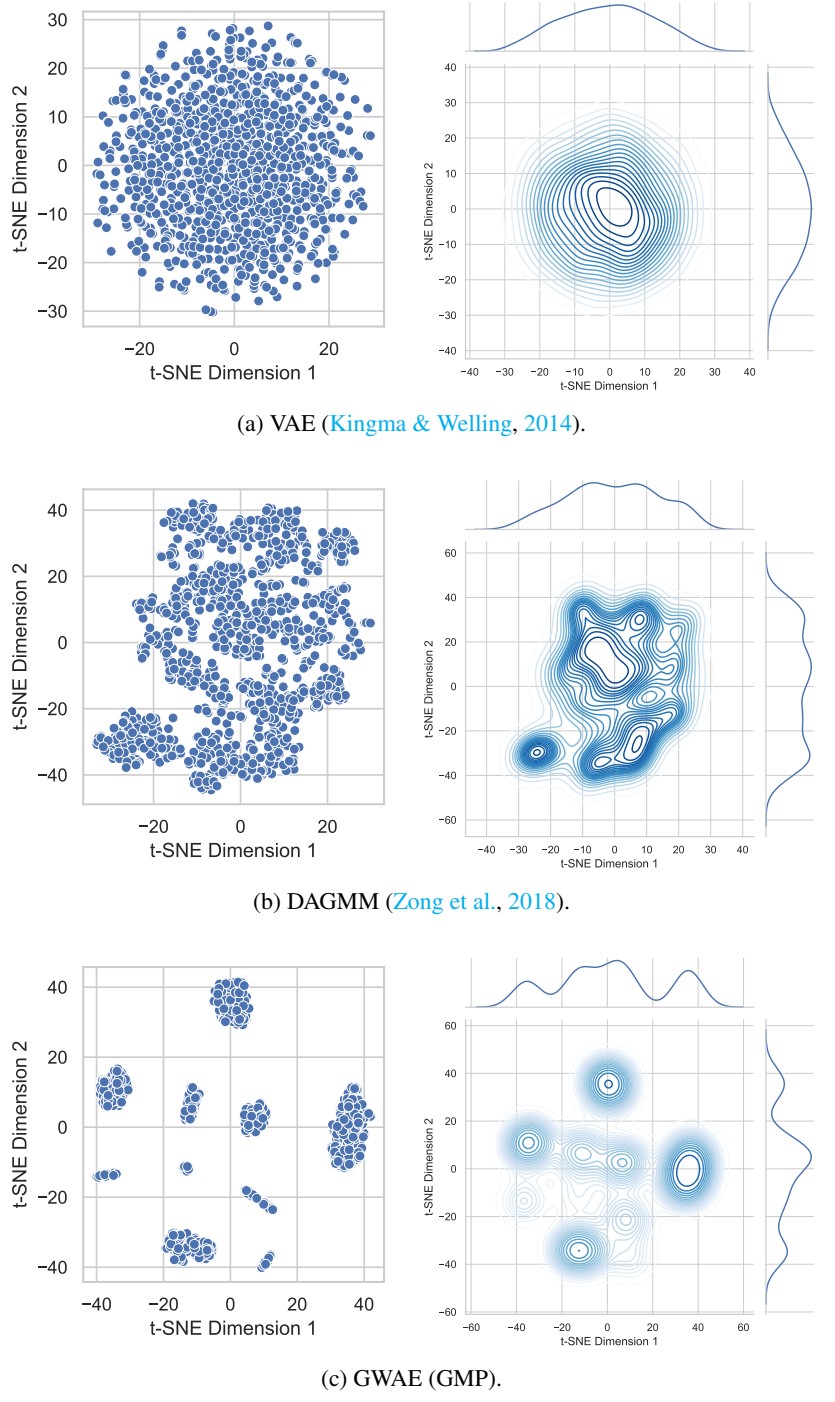

(a) VAE (Kingma & Welling, 2014).

(b) DAGMM (Zong et al., 2018).

(c) GWAE (GMP).

Figure 13: The t-SNE visualizations (van der Maaten & Hinton, 2008) for latent space samples $\mathbf{z} \sim \pi_{\boldsymbol{\theta}}(\mathbf{z})$ for the OoD detection in Fig. 3. The left plot presents the sampled points of the t-SNE embeddings, and the right one presents the kernel density estimation (KDE) of these embeddings. The sample size is equally 1,024 in each reported model.

