# OpenReview forum: "Gromov-Wasserstein Autoencoders"
_ICLR.cc/2023/Conference — ICLR 2023 poster_

### Official Review · Reviewer_y4yE · 2022-10-19

**Confidence:** 4
**Correctness:** 3
**Technical Novelty And Significance:** 3
**Empirical Novelty And Significance:** 2
**Recommendation:** 6

**Clarity, Quality, Novelty And Reproducibility:**

The method is sound, the assumptions are clear, and the empirical results clearly reproducible. The method is only marginally novel.

**Strength And Weaknesses:**

Pros:
1. The authors perform an extensive ablation study of the method which convinces me that the shown results are not statistical artifacts.
2. The method is based on a solid theory which is motivated well in the manuscript.
3. The derivations, although straightforward, are correct.


Cons:
1. The idea is a delta upon the large corpus of VAEs; importantly, it heavily builds upon WAE.
2. Even though the method shows promise in the considered datasets, I think that much more datasets are needed for the empirical evidence to be convincing.

**Summary Of The Paper:**

The paper is an extension of variational auto-encoders that uses a different type of metric for performing training. The suggested metric directly matches the latent and data distributions using the variational autoencoding scheme; this is based on the Gromov-Wasserstein (GW) metric between the trainable prior and given data distributions.



**Summary Of The Review:**

The method is interesting and sound, though marginally novel. The experiments show promise and depth, but we need more datasets for these to be convincing.

---

> ### Author Response · Authors · 2022-11-12
> **Response to Reviewer y4yE**
>
> Thank you very much for your informative comments. Here are our point-to-point clarifications for your concerns:
> - **The idea is a delta upon the large corpus of VAEs; importantly, it heavily builds upon WAE.**
> The novelty of the GWAE methodology lies in directly matching the latent and data spaces, which is the main difference from existing variational autoencoding methods including WAE. GWAE models directly optimize the latent space by minimizing the discrepancy between the latent and data space based on the Gromov-Wasserstein metric. The WAE objective is also OT-based; however, it measures the discrepancy between the $\mathcal{X}$-marginal of the generative model and the data distribution, and it optimizes the latent space $\mathcal{Z}$ via generative modeling. The GWAE optimization mentioned in Section 3.2.2 uses the WAE-based derivation in $\mathcal{L}\_{W}$, but it is rather a regularization to facilitate the estimation of the Gromov-Wasserstein metric in $\mathcal{L}\_{GW}$.
> - **Even though the method shows promise in the considered datasets, I think that much more datasets are needed for the empirical evidence to be convincing.**
> We agree with your comment that much more datasets would give richer evidence; however, considering the existing works, the set of datasets in the manuscript can be a standard set of datasets for benchmarking variational autoencoding models. For example, the WAE paper (Tolstikhin et al., 2018) uses MNIST and CelebA as the set of the experiments, the 2-Stage VAE paper (Dai & Wipf, 2019) MNIST, Fashion-MNIST, CIFAR-10 and CelebA, and the paper (Xiao et al., 2021) CIFAR-10, CelebA, CelebA-HQ, LSUN, and Stacked MNIST.
>
> #### References
> - [(Tolstikhin et al., 2018) Ilya Tolstikhin et al., Wasserstein auto-encoders. In ICLR, 2018.](https://openreview.net/forum?id=HkL7n1-0b)
> - [(Dai & Wipf, 2019) Bin Dai and David Wipf, Diagnosing and enhancing vae models. In ICLR, 2019.](https://openreview.net/forum?id=B1e0X3C9tQ)
> - [(Xiao et al., 2021) Zhisheng Xiao et al., VAEBM: A Symbiosis between Variational Autoencoders and Energy-based Models. In ICLR, 2021.](https://openreview.net/forum?id=5m3SEczOV8L)

---

### Official Review · Reviewer_JFYx · 2022-10-24

**Confidence:** 4
**Correctness:** 4
**Technical Novelty And Significance:** 3
**Empirical Novelty And Significance:** 3
**Recommendation:** 8

**Clarity, Quality, Novelty And Reproducibility:**

This work is clearly written and appears to me to novel. The results appear reproducible after a quick look at the submitted code.


**Strength And Weaknesses:**

[+] Convincing empirical evidence that GW metric is indeed estimated (Figure 1a).

[+] Good evidence that proposed method quantitatively improves disentanglement scores (Table 1).

[+] Comprehensive review of literature.

[-] Unclear empirical advantage of GWAE for sample generation task (Table 1). Prior work 2-stage VAE appears better.

[-] No discussion of computational cost or scalability of GW estimation. Isn't L_GW (Eq. 8) quadratically expensive?

[-] Only (relatively) small scale datasets are considered (CelebA, 3D Shapes, Omniglot). Is this related to the cost of estimating GW? More discussion is desired.



**Summary Of The Paper:**

A representation learning scheme is proposed using the Gromov-Wasserstein (GW) distance, a variant on optimal transport distances which allows comparison between distributions in different metric spaces. An auto-encoder based on the GW distance is proposed and evaluated on common image datasets.  Empirical tests on disentangling and clustering are made.


**Summary Of The Review:**

This work is a natural successor to many OT based generative modeling works existing in the literature. The empirical advantage of the proposed method appears limited with regards to generative quality measures, but does some improvements in disentanglement. I would like to see more discussion of cost of estimating GW. Nevertheless this work appears to me to be a worthwhile contribution and therefore I support acceptance.

---

> ### Author Response · Authors · 2022-11-12
> **Response to Reviewer JFYx**
>
> Thank you very much for your valuable comments. Here are our point-to-point clarifications and revisions for your concerns and questions:
> - **Unclear empirical advantage of GWAE for sample generation task (Table 2). Prior work 2-stage VAE appears better.**
> We agree with your comment that the prior work still has quantitatively better performance; however, as we mentioned in Section 4.4, we validated the distribution matching between the model $p_{\mathcal{D}}(\mathbf{x})$ and the data $p_{\mathbf{\theta}}(\mathbf{x})$ to study the substantial capture of the low-dimensional data manifold in latent codes. As GWAE performed the second best in Table 2, the similar performance to state-of-the-art existing methods suggests successful generative modeling for further applications to other meta-priors than two major meta-priors validated in Section 4.3.
> - **No discussion of computational cost or scalability of GW estimation. Isn't $\mathcal{L}_{GW}$ (Eq. 8) quadratically expensive?**
> For clarification, we added the explanation of the computational cost of GWAE in Appendix C.2. The cost of computing
>  $\mathcal{L}\_{GW}$ is $O(B^2)$ ($B$ being the batch size) for each epoch since estimating $\mathcal{L}_{GW}$ is performed in mini-batch stochastic descent. Since the number of mini-batch is $\lceil N/B \rceil$ ($N$ being the dataset size), the computational cost for each epoch is $O(NB)$. Hence, the GWAE methodology can be applied to large-scale datasets by choosing an appropriate batch size (e.g., 64 as mentioned in Appendix C.2).
> - **Only (relatively) small scale datasets are considered (CelebA, 3D Shapes, Omniglot). Is this related to the cost of estimating GW? More discussion is desired.**
> As mentioned above, the estimated GW $\mathcal{L}_{GW}$ is quadratic to the batch size, not to the dataset size, suggesting the applications to more large datasets. However, for comparisons, we used the datasets used in existing variational autoencoding works. MNIST and CelebA have been both used in the WAE paper (Tolstikhin et al., 2018) and the 2-Stage VAE paper (Dai & Wipf, 2019), and CIFAR-10 is used in the 2-Stage VAE paper (Dai & Wipf, 2019). CIFAR-10 and CelebA are also used in the paper (Xiao et al., 2021).
>
> #### References
> - [(Tolstikhin et al., 2018) Ilya Tolstikhin et al., Wasserstein auto-encoders. In ICLR, 2018.](https://openreview.net/forum?id=HkL7n1-0b)
> - [(Dai & Wipf, 2019) Bin Dai and David Wipf, Diagnosing and enhancing vae models. In ICLR, 2019.](https://openreview.net/forum?id=B1e0X3C9tQ)
> - [(Xiao et al., 2021) Zhisheng Xiao et al., VAEBM: A Symbiosis between Variational Autoencoders and Energy-based Models. In ICLR, 2021.](https://openreview.net/forum?id=5m3SEczOV8L)

---

> > ### Comment · Reviewer_JFYx · 2022-11-17
> > **Acknowledgement**
> >
> > Thanks to the authors for addressing my concerns. I have increased my score.

---

> > > ### Author Response · Authors · 2022-11-18
> > > **Thank you for your feedback**
> > >
> > > Thank you very much for your acknowledgement. Our paper has been improved thanks to your constructive feedback, and we are grateful that you have increased your score to reflect it.

---

### Official Review · Reviewer_ZoWi · 2022-10-24

**Confidence:** 4
**Correctness:** 3
**Technical Novelty And Significance:** 2
**Empirical Novelty And Significance:** 2
**Recommendation:** 5

**Clarity, Quality, Novelty And Reproducibility:**

This paper is well constructed and easy to understand. Experiments and their results are well described, but the proposed method seems to lack details. An autoencoder model that approximately minimizes the Gromov-Wasserstein distance is novel. Its ability of reconstructing the high-frequency components of hair seems novel, although need scrutization. It appears that the structure of the neural sampler of $Z$ is not included.

**Strength And Weaknesses:**

## Strengths

This paper proposes a method that can perform disentanglement and clustering tasks within a single model. The objective for the proposed model is intuitively derived. The performance of the proposed method is extensively demonstrated with many supporting experiments. The effect of each regularization term is experimentally confirmed.

## Weaknesses

1. Merging Eqs. (8) and (10) into the constraint $p_{\theta}(x,z) = q_{\phi}(x,z)$ needs a justification. The latter implies Eqs. (8) and (10), but the vice versa. I wonder if you can explain further about the advantages of using $\mathcal{L}_D$ in (11) compared to the maximum-mean discrepancy (MMD) between the aggregated posterior and the prior.

2. That the generated CelebA images preserves the details of hair is encouraging, but this may be a consequence of adding high-frequency noise to the generated smooth image. In fact, other parts of the faces exhibit spotty noise.



**Summary Of The Paper:**

This paper proposes an integrated VAE-based approach for representation learning by minimizing Gromov-Wasserstein distance between the metric-measure space of the data and latent vectors. Isometric encoding is obtained from the trained prior and their performances in meta-priors (e.g. disentanglement and clustering) are empirically demonstrated without additional penalty terms. In addition, the model has good generation performance although its goal is matching latent and data distributions.


**Summary Of The Review:**

An autoencoder model that approximately minimizes the Gromov-Wasserstein distance seems novel, although its derivation needs further justification. The proposed model is extensively compared with other models.

---

> ### Author Response · Authors · 2022-11-12
> **Response to Reviewer ZoWi**
>
> Thank you very much for your constructive and insightful comments. Here are our point-to-point clarifications and manuscript revisions for your concerns and questions:
> - **Merging Eqs. (8) and (10) into the constraint pθ(x,z) = qφ(x,z) needs a justification. ... compared to the maximum-mean discrepancy (MMD) between the aggregated posterior and the prior.**
> We merged the conditions Eqs. (8) and (10) into the joint constraint because the original variational autoencoding scheme aims to match the joint generative and inference models $p_{\mathbf{\theta}}(\mathbf{x},\mathbf{z})=q_{\mathbf{\phi}}(\mathbf{x},\mathbf{z})$.
> To experimentally validate its effect, we conduct the ablation study using $\mathcal{L}_D$ defined by the maximum-mean discrepancy (MMD), which measures distribution discrepancy widely used for comparing latent spaces in the VAE literature (Tolstikhin et al., 2018; Zhao et al., 2019).
> For a more fair comparison, we added the ablation study of $\mathcal{L}_D$ with a critic taking only $\mathbf{z}$ as its input in "GWAE (NP) w/ $\mathcal{Z}$-only critic" of Table 9, providing a clearer comparison between leaving the $\mathcal{Z}$-constraint in Eq.(10) unmerged and merging the $\mathcal{X}\times\mathcal{Z}$-constraints in Eqs. (8) and (10).
> - **That the generated CelebA images preserves ... other parts of the faces exhibit spotty noise.**
> By referring to the quantitative Peak Signal-to-Noise Ratio (PSNR) results shown in Table 2, we can validate whether GWAEs substantially reconstruct the original images. From the qualitative results in Figure 6 only, it appears to simply add high-frequency noises to blurry reconstructed images; however, Table 2 quantitatively shows the image reconstruction quality of GWAE and that the GWAE model was not simply adding visually impressive noises. Natural images typically include limited high-frequency activity as known in the image super-resolution literature (Park et al., 2003). It suggests that injecting high-frequency noises rather causes degrading reconstruction quality, and then that adding high-frequency noises generally results in low PSNR scores (also please note that PSNR is defined as the ratio of the signal power to the noise power). The GWAE model achieved the highest PSNR scores for reconstruction, which indicates that the high-frequency components of the reconstructed images are substantial signals, not simple perceptually preferable high-frequency noises. Hence, considering both quantitative and qualitative results, we could say that the GWAE model has substantially captured informative representations although there appear to be spotty noises in some images.
> - **It appears that the structure of the neural sampler of Z is not included.**
> Thank you for pointing out the omission. We added the network structures of samplers in Tables 6 and 7 (Appendix C.2).
> #### References
> - [(Park et al., 2003) Sung Cheol Park et al., Super-resolution image reconstruction: a technical overview, IEEE Signal Processing Magazine 20(3), pp. 21–36, 2003.](https://doi.org/10.1109/MSP.2003.1203207)
> - [(Tolstikhin et al., 2018) Ilya Tolstikhin et al., Wasserstein auto-encoders. In ICLR, 2018.](https://openreview.net/forum?id=HkL7n1-0b)
> - [(Zhao et al., 2019) Shengjia Zhao et al., InfoVAE: Balancing learning and inference in variational autoencoders. In AAAI, pp. 5885–5892, 2019.](https://doi.org/10.1609/aaai.v33i01.33015885)

---

> > ### Comment · Reviewer_ZoWi · 2022-11-18
> > **Acknowledgement**
> >
> > Thanks for the responses. I was expecting more compelling reasoning than just an ablation study for the use of (11) for $L_D$ instead of other divergences. To me, the choice of the penalties added to $\mathcal{L}_{GW}$ seems rather ad hoc. The remarkable aspect of WAE (Tolstikhin et al., 2018) is that it proves that the constrained optimization problem proposed in that paper indeed yields a Wasserstein distance, as long as the decoder is deterministic (so the validity of eqs. 9 and 10 in the submission is doubtful as they employ probabilistic decoder). The penalized form of this constrained problem is a principled relaxation of the constrained problem that allows gradient-based learning; if the penalty is strong enough, the solution converges to that of the constrained problem. In this sense the choice of the divergence for penalty is not super-critical in WAEs. However, there is no such guarantee in this submission. All the efforts for empirical evaluations are appreciated, but they need a principle.
> >
> > For these reasons I keep my score.

---

> > > ### Author Response · Authors · 2022-11-25
> > > **Further Clarification**
> > >
> > > Thank you very much for your further comment on this submission.
> > > For clarifying your concerns further, we would like to provide point-to-point explanations.
> > >
> > > - **I was expecting more compelling reasoning … All the efforts for empirical evaluations are appreciated, but they need a principle.**
> > > We respectfully agree with your comment on the importance of theoretical guarantees and principles; however, empirical studies play a key role in validating the applicability of GWAEs to our goal of capturing natural data.
> > > As we noted in Sections 1 and 3.2.1, we adopted assumptions and hypotheses for naturally observed data, which is potentially difficult to be theoretically proved (e.g., distributed on a manifold).
> > > The *justification* of such a methodology based on hypotheses needs to be supported by sufficiently deep and rich empirical evidence, which motivated us to conduct extensive experimental demonstrations and ablation studies.
> > > Whereas we empirically designed the generative model coupling assuming that a transport map by the Dirac decoder is sufficient for natural data, such a coupling family with a pushforward of a prior does not necessarily cover all couplings.
> > > To justify this Dirac decoder assumption, we experimentally studied the GWAE models in multiple meta-priors and generative modeling in Section 4 and Appendix C, which indeed showed the effectiveness of GWAEs for learning representations for high-dimensional natural data.
> > > For example, Figs. 1 and 12 show the difference in the GW estimator’s behavior between prior family choices, suggesting the effect of the meta-prior choice in representation learning.
> > > - **(so the validity of eqs. 9 and 10 in the submission is doubtful as they employ probabilistic decoder)**
> > > Please also note that we have used Dirac decoders as mentioned in the last sentence of Section 3.2.1 and that the $\mathcal{X}$-marginal constraint by Eqs. (9) and (10) is still valid in terms of the WAE methodology.
> > >
> > > In summary, the justification of the GWAE methodology requires empirical studies because it starts from hypotheses and meta-priors about (almost) naturally observed high-dimensional data, which is to be confirmed by experimental observations.
> > >
> > > We hope that these explanations can help clarify your concern about our hypothesis verification process.
> > > If you have further concerns and questions, we would be glad to address them before the end of the discussion stage.

---

> > > > ### Comment · Reviewer_ZoWi · 2022-12-04
> > > > **Justification**
> > > >
> > > > I am not challenging the manifold hypothesis. What I would like to see under the name of "Gromov-Wasserstein Autoencoder" is, a theoretical guarantee that under some idealization (global optimization and large enough penalties, etc.)  the GWAE objective coincides with the Gromov-Wasserstein distance. The WAE achieves this goal for the Wasserstein distance; the proposed GWAE does not for the advertised distance.

---

> > > > > ### Author Response · Authors · 2022-12-08
> > > > > **Response**
> > > > >
> > > > > Thank you for your further response.
> > > > > We agree with the concern that the title consisting of only *“Gromov-Wasserstein Autoencoders”* implies an excessively general claim as if the paper contained theoretical contributions rather than empirical studies.
> > > > > In the paper, a novel representation learning framework is proposed and its empirical effectiveness is shown in several aspects including the match of the estimated and empirical GW metric (Section 4.2).
> > > > > To deal with the gap of expected contributions, we will replace the current paper title with *“Meta-Prior-Based Representation Learning via Gromov-Wasserstein Autoencoders”* in a camera-ready version to focus on a representation learning framework supported on rich empirical evidence as mentioned in the paper rather than the theoretical contribution to autoencoders.

---

### Official Review · Reviewer_PxfM · 2022-10-25

**Confidence:** 4
**Correctness:** 3
**Technical Novelty And Significance:** 3
**Empirical Novelty And Significance:** 3
**Recommendation:** 8

**Clarity, Quality, Novelty And Reproducibility:**

The paper is relatively easy to follow, though somewhat inconvenient due to the large amount of content in the supplementary material. The approach seems novel and properly justified in light of the existing literature, though innovation is somewhat difficult to assess considering the complexity of the proposed objective (for instance ablation without L_GW is not considered). The authors provide details of the data, model parameter, optimization details and source code that should in principle facilitate reproducibility.

**Strength And Weaknesses:**

The authors present a new perspective of using GW for the construction of flexible variational autoencoders. For the most part the proposed approach is well motivated and properly justified, though requiring three auxiliary and complementary (based on the ablation studies) loss functions. Though mostly presented in the supplementary material, the authors present a thorough overview of the variational autoencoding landscape.

The work has some key weaknesses:

The justification for \rho=1 citing alleviating outlier effects is not necessarily convincing, more so when in (9) they adopt a \zeta=2 reconstruction loss, so why are outliers a concern in (7) but not in (9)?

Though the ablation study is a welcome addition to the experiments as they demonstrate the contribution of L_W, L_D and R_H to (13), it will be important to include results without L_GW.

The results in Figure 2 though impressive at first raise some questions: i) it seems that both VAE and GWAE separate (disentangle) object hues in 2 out of the 16 dimensions considered, and ii) even though it seems that the x-axis in Figure 2c captures the object hue, either it does not preserve the order (see color map) or there is no ordering in the hue, in which case one dimension captures the hue, but one wonders how consistently across runs and model hyperparameters.

In Table 2, it is not clear how biased are the results considering that hyperparameters were selected to optimize DCI-C that is used as performance metric and happens to show the largest difference with existing methods.

From Table 2 it is not clear why the authors could not obtain results for WAE considering the closeness with the proposed approach. Further, the table seems to indicate that FID was taken from the original papers but not PSNR?

The results for clustering structure are interesting, however, are probably not the best way to showcase the clustering capabilities of the prior used in the proposed approach. In fact, as the authors show, a relatively simple VAE lacking a clustering prior achieves near perfect outlier detection.

The authors do not discuss why for Section 2 they use NP exclusively. It will be interesting to see results for GW estimation and minimization using FNP.

Minor points:

The marginal p_\theta(x) in (8) is not defined.

**Summary Of The Paper:**

The authors propose Gromov-Wasserstein Autoencoders (GWAE). The application of the GW metric allows to match data distributions with a given learnable prior, even when the distributions lie in spaces of different dimensions, thus allowing matching between latent and data space (in equation 4). Further, the GW objective allows rich meta-priors for flexible representation learning. Experiments on disentanglement and clustering using benchmark datasets (CelebA, MNIST and 3D Shapes) demonstrate the capabilities of the proposed approach.

**Summary Of The Review:**

The authors offer an interesting and seemingly flexible approach to optimize variational autoencoders by leveraging the GW metric, which allows direct matching between observed and latent spaces, however, not without the help of complementary objective functions, specifically, marginal matching given the approximate posterior (WAE), critic-based joint distribution matching (KR), and entropy regularization to alleviate collapsing.

---

> ### Author Response · Authors · 2022-11-12
> **Response to Reviewer PxfM (1/3)**
>
> (The reference list is shown in the last (third) comment.)
>
> Thank you very much for your detailed and insightful comments. Here are our point-to-point clarifications and revisions for your key concerns and questions:
>
> - **The justification for \rho=1 citing alleviating outlier effects is not necessarily convincing, ... so why are outliers a concern in (7) but not in (9)?**
> We agree with your comment and we have revised the manuscript for clarification. We set the GW order ρ=1 for the training stability of GW minimization, in which $\rho>1$ makes training difficult due to outlier samples with a large gap between $d_\mathcal{X}$ and $d_\mathcal{Z}$. We conducted additional ablation experiments in Appendix C.8 and Table 9 (Ablation Study) to compare $\rho=1$ and $\rho=\xi(=2)$.
> - **Though the ablation study is a welcome addition ..., it will be important to include results without L_GW.**
> The ablation of LGW could give the implication of its necessity; however, such an ablation study actually means comparisons with existing variational autoencoding models, having been conducted in the experiments of Table 2. The key difference of GWAE models from existing variational autoencoding models is using the GW metric estimated by the loss LGW, for which the other three regularization terms could be potentially omittable and it has required these ablation studies. Considering this key difference, the ablation of LGW would mean building existing VAE-based models such as WAE and ALI, which have been compared in Table 2.
> Please also refer to Appendix A for the details of such existing methods.
> - **The results in Figure 2 though impressive at first raise some questions:**
>     - **i) it seems that both VAE and GWAE separate (disentangle) object hues in 2 out of the 16 dimensions considered,**
>     We agree with your comment that both models capture hues within two of 16 dimensions; however, disentanglement is commonly defined as a representation in which a single representational variable corresponds to a single generative factor, as in the disentangled representation learning literature (Bengio et al., 2013; Higgins et al., 2017; Locatello et al., 2019). In this respect, it would be still important to compare two of the VAE latent variables and only one of the GWAE latent variables for representing a single factor, object hue. Such singleness was also quantitatively observed in Table 1 as DCI-C, which measures the degree to which each generative factor is represented by a single latent variable (Eastwood & Williams, 2018).
>     - **and ii) even though it seems that ... no ordering in the hue, in which case one dimension captures the hue,**
>     We agree with your concern that the learned representation does not preserve hue order and starts from the left with the non-zero value; however, these results would be a successful one because the color hue is in the color wheel and has intrinsically one degree of freedom in angle offset. We can consider some orders by cutting open the color wheel from an offset angle, but there would be no unique offset intrinsically determined because the hue is a periodic factor.
>     - **but one wonders how consistently across runs and model hyperparameters.**
>     Since disentanglement performance tends to be sensitive to hyperparameters and runs, model selection is commonly conducted via validation as a standard procedure of disentangled representation learning (Higgins et al., 2020). Hence, we reported one instance of latent representations selected by hyperparameter tuning using the validation set. For fair comparisons, we have spent substantially the same amount of computational costs ($\approx$20 hours in a single GPU) for the hyperparameter tuning of each model, and hence presented one of the hyperparameter setting sets selected by validation, not across runs and hyperparameters.

---

> ### Author Response · Authors · 2022-11-12
> **Response to Reviewer PxfM (2/3)**
>
> (The reference list is shown in the last (third) comment.)
>
> - **In Table 2, it is not clear how biased ... happens to show the largest difference with existing methods.**
> We agree with your concern that both hyperparameter tuning and evaluations are in the same performance metric DCI-C without clear justification; we revised the disentanglement paragraph in Section 4.3 to clarify the reason to evaluate the singleness of representation for each latent factor. Since the ground-truth factor size is often unknown in practical applications, it is required to set a large latent size to avoid the shortage of informativeness due to too small a latent size. Hence, in this experiment of Table 2, the latent size of L=16 is larger than the ground-truth factor size six (Burgess & Kim, 2018). In such a situation, ten latent variables should be collapsed (uninformative) even in the ideal one-to-one correspondence of latent variables and generative factors. The DCI-C metric is suitable to evaluate such one-to-one correspondence in the subspace of the latent space because DCI-C measures the representation singleness of each generative factor. For this reason, we selected model hyperparameters by the validation DCI-C and reported the test DCI-C value.
> - **From Table 2 it is not clear why the authors could not obtain results for WAE considering the closeness with the proposed approach.**
> Since the model settings of existing variational autoencoding works vary from paper to paper (e.g., preprocessing procedures, network architectures), we could not reproduce the original FID values in our settings. The papers we cited adopt different settings and provide different scores for the same method WAE: 55 in the original paper (Tolstikhin et al., 2018), and 62.9 in (Dai & Wipf, 2019). Please note that the values of Table 2 are still comparable because the settings in the calculation of FID scores are the same as the previous works, i.e., using Inception-V3 pre-trained by ImageNet (in Appendix C.3.2).
> - **Further, the table seems to indicate that FID was taken from the original papers but not PSNR?**
> We calculated PSNR scores because the original papers do not provide autoencoding experiments. Please note that It would not affect the consistency of Table 2 since we conducted the hyperparameter tuning and model selection separately for generative modeling experiments (FID) and autoencoding ones (PSNR).
> - **The results for clustering structure are interesting, however, ... a relatively simple VAE lacking a clustering prior achieves near perfect outlier detection.**
> We agree with your comment that the capture of clusters is not clear from the fact that VAE without cluster structure has nearly perfect performance, presumably owing to the MNIST dataset having a simple background and format.
> In Appendix C.10 and Figure 13, we added the scatter plots of their latent spaces to compare the existence of clusters more clearly.
> - **The authors do not discuss why for Section 2 they use NP exclusively. It will be interesting to see results for GW estimation and minimization using FNP.**
> Thank you for your insightful suggestion on selecting a prior family. We have solely used NP to present the model behavior of general and unconstrained prior cases; therefore, we added the experimental results for GW estimation and minimization using FNP in Appendix C.9. Presumably due to the mismatch of disentanglement meta-prior for the MNIST dataset, the GW metric was not more precisely estimated and the isometry was not tighter than NP. Please also note that FNP is a subset of NP because FNP is built by constraining NP to be factorized.
> - **The marginal $p_{\mathbf{\theta}}(\mathbf{x})$ in (8) is not defined.**
> Thank you for pointing out the omission. We added the definition of the marginal $p_{\mathbf{\theta}}(\mathbf{x})$ right after Eq. (8).

---

> ### Author Response · Authors · 2022-11-12
> **Response to Reviewer PxfM (3/3)**
>
> (The first and second pages are below; please first refer to these comments.)
>
> ### References
> - [(Bengio et al., 2013) Yoshua Bengio et al., Representation learning: A review and new perspectives. IEEE Transactions on Pattern Analysis and Machine Intelligence, 35(8):1798–1828, 2013.](https://doi.org/10.1109/TPAMI.2013.50)
> - [(Higgins et al., 2017) Irina Higgins et al., Matthew Botvinick, Shakir Mohamed, and Alexander Lerchner. β-VAE: Learning basic visual concepts with a constrained variational framework. In ICLR, 2017.](https://openreview.net/forum?id=Sy2fzU9gl)
> - [(Eastwood & Williams, 2018) Cian Eastwood and Christopher K. I. Williams. A framework for the quantitative evaluation of disentangled representations. In ICLR, 2018.](https://openreview.net/forum?id=By-7dz-AZ)
> - [(Burgess & Kim, 2018) Chris Burgess and Hyunjik Kim, 3D Shapes Dataset, 2018.](https://github.com/deepmind/3d-shapes)
> - [(Tolstikhin et al., 2018) Ilya Tolstikhin et al., Wasserstein auto-encoders. In ICLR, 2018.](https://openreview.net/forum?id=HkL7n1-0b)
> - [(Dai & Wipf, 2019) Bin Dai and David Wipf, Diagnosing and enhancing vae models. In ICLR, 2019.](https://openreview.net/forum?id=B1e0X3C9tQ)
> - [(Locatello et al., 2019) Francesco Locatello et al., Challenging common assumptions in the unsupervised learning of disentangled representations. In ICML, pp. 4114–4124, 2019.](http://proceedings.mlr.press/v97/locatello19a.html)
> - [(Higgins et al., 2020) Irina Higgins et al., Unsupervised Model Selection for Variational Disentangled Representation Learning, In ICLR, 2020.](https://openreview.net/forum?id=SyxL2TNtvr)

---

### Author Response · Authors · 2022-11-12
**Response to Reviewers**

We are grateful for your constructive feedback, which is very helpful for the revisions to improve and fortify the manuscript. The main points of the revisions are listed below:

- We added further explanations for the experimental settings in the disentanglement part of Section 4.3 (Reviewer PxfM).
- In Appendix C.8, to fortify the ablation study, we added the GWAE model replacing the merged sufficient condition LD with a $\mathbf{z}$-only critic based on the $\mathcal{Z}$-condition of Eq. (10) only (Reviewer ZoWi).
- We added further ablation studies on the merged sufficient condition in Appendix C.8 (Reviewer PxfM).
In Appendix C.8, we added the ablation study of the $\rho=1$ setting, additionally experimenting in the $\rho=\xi$ setting that appears to be intuitively natural but practically causes performance degradation (Reviewer PxfM).
- We added Appendix C.9 to show the GW estimation and minimization of the GWAE model with FNP (Reviewer PxfM).
- We added Appendix C.10 to show the latent space in the Out-of-Distribution detection task of Figure 3 (Reviewer PxfM).

We would greatly appreciate it if you could refer to our responses below and revisions above. If there are further questions/comments/suggestions, we would be happy to address them in the discussion period.

---

### Comment · Area_Chair_7NUY · 2022-12-05
**Question**

Dear authors,

 Could you please explain the relationship between the proposed method and the relation regularized autoencoder method, which follows the Tolstikhin et al. 2018's work?
#http://proceedings.mlr.press/v119/xu20e/xu20e.pdf (You cited this paper in the main body)

 Thanks,

AC

---

> ### Author Response · Authors · 2022-12-08
> **Response to Question**
>
> Dear Area Chair 7NUY,
>
> The main difference between our GWAE and the relational regularized autoencoder (RAE) in Xu et al. (2020) is the motivation of introducing the Gromov-Wasserstein (GW) metric and hence the spaces compared by GW.
> RAE introduces the fused GW metric between the aggregated posterior $q_{{\phi}}(\mathbf{z})$ and the latent prior $p_{{\theta}}(\mathbf{z})$ as the *regularization divergence* to fortify the WAE constraint $p_{{\theta}}(\mathbf{z})=q_{{\phi}}(\mathbf{z})$ (Tolstikhin et al., 2018) for generative modeling.
> Our GWAE introduces the GW metric between the data distribution $p_{\mathcal{D}}(\mathbf{x})$ and the latent prior $p_{{\theta}}(\mathbf{z})$ as an *objective* to explicitly learn representations in the latent variables $\mathbf{z} \sim p_{{\theta}}(\mathbf{z})$ based on data and meta-priors.
> It aims at a more direct method to learn representations, rather than representation learning via generative modeling.
> In summary, our GWAE method aims to introduce the objective to directly match the latent prior $p_{{\theta}}(\mathbf{z})$ and the data distribution  $p_{\mathcal{D}}(\mathbf{x})$ for learning representations, while the RAE method aims the regularization to match the latent prior $p_{{\theta}}(\mathbf{z})$ and the aggregated posterior $q_{{\phi}}(\mathbf{z})$ for WAE-based generative modeling.
> #### References
> - [(Tolstikhin et al., 2018) Ilya Tolstikhin et al., Wasserstein auto-encoders. In ICLR, 2018.](https://openreview.net/forum?id=HkL7n1-0b.)
> - [(Xu et al., 2020) Hongteng Xu et al.,  Learning Autoencoders with Relational Regularization. In ICML 2020.](https://proceedings.mlr.press/v119/xu20e.html)

---

### Author Response · Authors · 2023-02-24
**Camera Ready Revision**

Greetings,

Thank you very much for accepting our paper at the ICLR2023 conference, and we would like to leave some clarifications about camera ready revision for readers.

**Paper Title.**
Although we indicated in the rebuttal that we will change the paper title, we have decided to maintain the original title “*Gromov-Wasserstein Autoencoders*,” reconsidering Reviewer ZoWi’s comment and the Program Chairs’ recommendation.
Reviewer ZoWi mentioned the name of our proposed method in [the comment “Justification,”](https://openreview.net/forum?id=sbS10BCtc7&noteId=c7johxWh2HU) but we incorrectly interpreted it as he was referring to the title of the paper.
Owing to this misunderstanding, changing the paper title was not a very accurate reply, which may rather make the focus of our paper unclear.
Program Chairs have also mentioned the proposed formulation as its novelty and our method as based on the Gromov-Wasserstein loss.
The changed title focusing on meta-priors does not very accurately convey the novelty, which may confuse readers.
For these reasons, we decided not to change the title of this paper and will adopt the title of the initial submission in a camera ready revision.

**Program Chairs’ Recommendation.**
Following the Program Chairs’ recommendation, we have also added the following contents to this paper:
- We added the description of the relational regularized autoencoder (RAE) in Appendix A.5. The objective of RAE includes the GW metric as a regularization term, which measures the discrepancy between the latent spaces.
- We added the quantitative results of RAEs (Xu et al., 2020) in Table 2 for comparison with a recent variational autoencoding method.

**Code.**
Since the author list is now made public, we have released the code of the proposed method, and it is available online at https://github.com/ganmodokix/gwae. We added this repository URL to the Reproducibility Statement section of the camera ready version.

We would like to apologize for the misunderstanding that we made in the rebuttal to Reviewer ZoWi, and we respectfully thank the Reviewers, Area Chair, and Program Chairs for improving our paper through their comments and discussions.

Sincerely,

The authors

### Reference
(Xu et al., 2020) [Hongteng Xu et al., Learning Autoencoders with Relational Regularization. In ICML 2020.](https://proceedings.mlr.press/v119/xu20e.html)

---

### Decision · Program_Chairs · 2023-01-20

**Decision:**

Accept: poster

**Justification For Why Not Higher Score:**

The paper has a score larger than borderline. However, there exist some concerns in the paper:
1. The discussion with recent work about autoencoder with Gromov-Wasserstein loss is not adequately discussed. (e.g., http://proceedings.mlr.press/v119/xu20e/xu20e.pdf )
2. The proposed method only compares relatively old methods.

**Justification For Why Not Lower Score:**

At least, the proposed formulation is new and different from the previous work. Moreover, a reviewer is excited to publish the paper. So, it is good to be in the venue.

However, there exist some concerns in the paper:
1. The discussion with recent work about autoencoder with Gromov-Wasserstein loss is not adequately discussed. (e.g., http://proceedings.mlr.press/v119/xu20e/xu20e.pdf )
2. The proposed method only compares relatively old methods.

So, I am OK with the paper is rejected if there is not enough space for presentation.

**Metareview: Summary, Strengths And Weaknesses:**

In this paper, the authors propose a new auto-encoder based on Gromov-Wasserstein loss, where the proposed method is based on WAE (Tolstikhin et al. 2018). The proposed approach is interesting and practically works well. Overall, the reviewers agree to accept the paper; I will also vote for acceptance.

However, there exist some concerns in the paper:
1. The discussion with recent work about autoencoder with Gromov-Wasserstein loss is not adequately discussed. (e.g., http://proceedings.mlr.press/v119/xu20e/xu20e.pdf )
2. The proposed method only compares relatively old methods.

I strongly encourage authors to address these issues in the final version.

**Note From Pc:**

if the above contains the word "oral" or "spotlight" please see: "oral" presentation means -> notable-top-5% and "spotlight" means -> notable-top-25%. As stated in our emails, we are disassociating presentation type from AC recommendations